# Replicable Uniformity Testing

**Sihan Liu**
University of California San Diego
La Jolla, CA
sil046@ucsd.edu

**Christopher Ye**
University of California San Diego
La Jolla, CA
czye@ucsd.edu

## Abstract

Uniformity testing is arguably one of the most fundamental distribution testing problems. Given sample access to an unknown distribution $\mathbf{p}$ on $[n]$, one must decide if $\mathbf{p}$ is uniform or $\varepsilon$-far from uniform (in total variation distance). A long line of work established that uniformity testing has sample complexity $\Theta(\sqrt{n}\varepsilon^{-2})$. However, when the input distribution is neither uniform nor far from uniform, known algorithms may have highly non-replicable behavior. Consequently, if these algorithms are applied in scientific studies, they may lead to contradictory results that erode public trust in science.

In this work, we revisit uniformity testing under the framework of algorithmic replicability [STOC '22], requiring the algorithm to be replicable under arbitrary distributions. While replicability typically incurs a $\rho^{-2}$ factor overhead in sample complexity, we obtain a replicable uniformity tester using only $\tilde{O}(\sqrt{n}\varepsilon^{-2}\rho^{-1})$ samples. To our knowledge, this is the first replicable learning algorithm with (nearly) linear dependence on $\rho$.

Lastly, we consider a class of "symmetric" algorithms [FOCS '00] whose outputs are invariant under relabeling of the domain $[n]$, which includes all existing uniformity testers (including ours). For this natural class of algorithms, we prove a nearly matching sample complexity lower bound for replicable uniformity testing.

## 1 Introduction

Distribution property testing (see [43, 9, 12] for surveys of the field), originated from statistical hypothesis testing [39, 36], aims at testing whether an unknown distribution satisfies a certain property or is significantly "far" from satisfying the property given sample access to the distribution.

After the pioneering early works that formulate the field from a TCS perspective [5, 4], a number of works have achieved progress on testing a wide range of properties [6, 21, 11, 22, 10, 19]. Within the field, uniformity testing [28] is arguably one of the most fundamental distribution testing problems: Given sample access to some unknown distribution $\mathbf{p}$ on $[n] = \{1, \cdots, n\}$, one tries to decide whether $\mathbf{p}$ is uniform or far from being uniform.

When the unknown distribution is promised to be either uniform or at least $\varepsilon$-far from being uniform in total variation (TV) distance, a line of work in the field [28, 40, 47, 20, 1, 16] has led to efficient testers that achieve information theoretically optimal sample complexity, i.e., $\Theta(\sqrt{n}/\varepsilon^2)$. However, the testers are only guaranteed to provide reliable answers when $\mathbf{p}$ fulfills the input promise — $\mathbf{p}$ is either $U_n$ or far from it. In practical scenarios, due to a number of reasons such as model misspecification, or inaccurate measurements, such promises are rarely guaranteed [13]. Since the tester no longer has a correctness constraint to adhere to, it may have arbitrarily volatile behavior. Ideally, we hope for algorithmic stability for all distributions, not only on those fulfilling the promises. Consider the scenario where two group of scientists, as an intermediate step of some scientific study, are trying to independently test the uniformity of some ground-truth distribution $\mathbf{p}^*$. If the groups

38th Conference on Neural Information Processing Systems (NeurIPS 2024).

reach inconsistent conclusions in the end (as noted in [31, 32, 7], such replicability issues are fairly common in science), the public might naturally interpret the inconsistency to be caused by procedural or human errors on the part of one (or both) team(s), therefore eroding public trust in the scientific community [37, 44, 3, 42]. However, even when both teams follow the experimental procedure precisely, inconsistencies could still arise simply due to sample variance in the case where $\mathbf{p}^*$ is neither uniform nor far from uniform. By ensuring that the algorithm is *replicable*, we can effectively rule out sample variance as a cause of inconsistency.

In the work of [30], a formal definition of replicability for learning algorithms has been proposed to mitigate non-replicability caused by statistical methods. Specifically, replicable learning algorithms are required to give identical outputs with high probability in two different runs where it is given sample access to some common, but potentially adversarially chosen data distribution.

**Definition 1.1** (Replicability [30]). *A randomized algorithm $\mathcal{A} : \mathcal{X}^n \mapsto \mathcal{Y}$ is $\rho$-replicable if for all distributions $\mathbf{p}$ on $\mathcal{X}$, $\Pr_{r,T,T'} (\mathcal{A}(T; r) = \mathcal{A}(T'; r)) \geq 1 - \rho$, where $T, T'$ are i.i.d. samples taken from $\mathbf{p}$ and $r$ denotes the internal randomness of the algorithm $\mathcal{A}$.*

A number of works have explored the connection between replicability and other algorithmic stability notions [8, 34, 15, 38, 23], and shown efficient replicable algorithms for a wide range of machine learning tasks [8, 34, 24, 25, 35, 26, 33]. In the context of uniformity testing, if we can design testers that conform to the above replicability requirement, consistencies of their outputs can be guaranteed even when there are no promises on the data distribution. This then motivates the study of *replicable uniformity testing*.

**Definition 1.2** (Replicable Uniformity Testing). *Let $n \in \mathbb{Z}_+$, and $\varepsilon, \rho \in (0, 1/2)$. A randomized algorithm $\mathcal{A}$, given sample access to some distribution $\mathbf{p}$ on $[n]$, is said to solve $(n, \varepsilon, \rho)$- replicable uniformity testing if $\mathcal{A}$ is $\rho$-replicable and it satisfies the following: (1) If $\mathbf{p}$ is uniform, $\mathcal{A}$ should accept with probability at least $1 - \rho$. [1] (2) If $\mathbf{p}$ is $\varepsilon$-far from the uniform distribution in TV distance, $\mathcal{A}$ should reject with probability at least $1 - \rho$.*

As observed in [30], learning algorithms usually incurs additional sample complexity overhead in the replicability parameter $\rho$ compared to their non-replicable counterpart. In this work, we characterize the sample complexity of replicable uniformity testing up to polylogarithmic factors in all relevant parameters $n, \varepsilon, \rho$ (under mild assumptions on the testers).

## 1.1 Relationship with Tolerant Testing

An alternative approach to address the stringency of the promises in the formulation of uniformity testing is the concept of *tolerant testing* [41]. At a high level, given some $0 < \xi < \varepsilon$, the unknown distribution is now relaxed to be either $\varepsilon$ far from $U_n$ or $\xi$ close to $U_n$ in TV distance. The tester is then required to reject in the former case but accept in the latter. As shown in [45, 46, 48, 13], assuming that $\varepsilon$ is some constant [2], the sample complexity of tolerant testing quickly grows from strongly sublinear, i.e., $\Theta(\sqrt{n})$, to barely sublinear, i.e, $\Theta(n/\log n)$ as $\xi$ increases from 0 to $\varepsilon/2$. In replicable uniformity testing the testers are not required to accept or reject for intermediate distributions $\mathbf{p}$, i.e., $\mathbf{p}$ such that $0 < \mathrm{TV}(\mathbf{p}, U_n) < \varepsilon$. Instead, for all possible distributions $\mathbf{p}$, the testers are only required to give *replicable* answers (with high probability). While we can construct an $(n, \varepsilon', \rho)$-replicable uniformity testing algorithm using tolerant testing by randomly sampling some threshold $r \in (0, \varepsilon')$ and performing tolerant testing with $\xi = r - \rho\varepsilon'$ and $\varepsilon = r + \rho\varepsilon'$, the sample complexity of such an approach will be barely sublinear in $n$ even for constant $\rho$ as discussed above. Notably, the replicable algorithm in our work has a sample complexity that remains strongly sublinear in $n$.

## 1.2 Our Results

When $n = 2$, uniformity testing amounts to distinguishing a fair coin from an $\varepsilon$-biased coin. It is well known that the task requires $\Theta(\varepsilon^{-2})$ samples without the replicable requirement. When the tester is required to be $\rho$-replicable, [30] shows that $\tilde{\Theta}(\varepsilon^{-2}\rho^{-2})$ many samples are necessary and sufficient

---

[1] In this work, we do not focus on the dependency on the failure probability of the tester. The common formulation is that $\mathcal{A}$ should succeed with some large constant probability, i.e., 2/3. However, if $\mathcal{A}$ is at the same time required to be replicable with probability at least $1 - \rho$, one can see that $\mathcal{A}$ must also be correct with probability at least $1 - \rho$.

[2] See [13] for the dependency on $\varepsilon$ and $\xi$ for the full landscape of the problem.

for replicable coin testing, demonstrating a quadratic blowup in $\rho$ compared to the non-replicable counterpart of the problem. For large $n$, the sample complexity of non-replicable uniformity testing has been resolved after a long line of work [28, 40, 47], and shown to be $\Theta(\sqrt{n}\varepsilon^{-2})$. Following the pattern, one naturally expect that replicable uniformity testing would have sample complexity $\tilde{\Theta}(\sqrt{n}\rho^{-2}\varepsilon^{-2})$. In fact, this is exactly the sample complexity reached if one views the outcome of an optimal non-replicable uniformity tester as a coin flip, and uses an optimal replicable coin tester to convert the given uniformity tester into a replicable one.

Somewhat surprisingly, we show that the sample complexity from this blackbox reduction is sub-optimal for replicable uniformity testing — it is possible to additionally shave one $\rho$ factor and make the dominating term's dependency on $\rho$ *linear*. To our knowledge, this is the first replicable algorithm that has nearly linear dependence on the replicability parameter in the dominating term.

**Theorem 1.3** (Replicable Uniformity Testing Upper Bound)**.** *Let $n \in \mathbb{Z}_+$, $\varepsilon, \rho \in (0, 1/2)$. Algorithm 1 solves $(n, \varepsilon, \rho)$-replicable uniformity testing with sample complexity $\tilde{O}\left(\frac{\sqrt{n}}{\varepsilon^2\rho} + \frac{1}{\rho^2\varepsilon^2}\right)$.*[3]

**Remark 1.4.** *[27] showed that the more general problem of identity testing, i.e., testing whether an unknown distribution $\mathbf{p}$ is equal or far from some known distribution $\mathbf{q}$ with explicit description, can be reduced to uniformity testing with only a constant factor loss in sample complexity. It is not hard to verify that this reduction preserves replicability. This immediately implies a replicable identity tester with the same asymptotic sample complexity (see Appendix C.3 for more detail).*

As our second result, we provide a nearly matching sample complexity lower bound for a natural class of testers whose outputs are invariant under relabeling of the domain elements $[n]$. The class of testers are commonly referred as *Symmetric Algorithms*, and first studied in the work of [5].

**Definition 1.5** (Symmetric Algorithms, Definition 13 of [5])**.** *Let $f : [n]^{\times m} \mapsto \{0, 1\}$ be a binary function. We say that $f$ is symmetric if for any sample set $(x_1, \cdots x_m) \in [n]^{\times m}$ and any permutation $\pi \in S_n$, we have that $f(x_1, \cdots x_m) = f(\pi(x_1), \cdots, \pi(x_m))$. We say an algorithm $\mathcal{A}(; r)$ is symmetric if it computes some symmetric function $f_r$ under any fixed random seed $r$.*

Without the replicability requirement, it can be shown that assuming the algorithm is symmetric is without loss of generality for uniformity testing. This is due to a simple observation that uniformity itself is a symmetric property: if $p$ is uniform or far from the uniform distribution, the property is preserved even if we permute the labels of the elements within $[n]$. Consequently, all known optimal uniformity testers, including our replicable uniformity tester, are indeed symmetric algorithms. For this natural class of testers, we show that the sample complexity achieved is essentially optimal (up to polylogarithmic factors).

**Theorem 1.6** (Symmetric Testers Lower Bound)**.** *Let $n \in \mathbb{Z}_+$, $\varepsilon, \rho \in (0, 1/2)$. Any symmetric algorithm solving the $(n, \varepsilon, \rho)$-replicable uniformity testing problem requires $\tilde{\Omega}\left(\frac{\sqrt{n}}{\varepsilon^2\rho} + \frac{1}{\rho^2\varepsilon^2}\right)$ samples.*

Unfortunately, when replicability is concerned, it is unclear whether we can still assume that the optimal algorithm is symmetric. Whether the above lower bound holds for all algorithms is left as one of the main open questions of this work.

## 1.3 Limitations, Discussion, and Future Work

In this paper, we present a replicable algorithm for uniformity testing using $\tilde{O}(\sqrt{n}\varepsilon^{-2}\rho^{-1} + \varepsilon^{-2}\rho^{-2})$ samples. We provide a matching lower bound for a natural class of *symmetric* uniformity testers — algorithms that essentially consider only the *frequency* of each element without discriminating between distinct labels. Since all known uniformity testing algorithms are symmetric, we tend to believe that the lower bound can be established *unconditionally* and leave it as an open question. We discuss some issues in generalizing our current approach to realize this goal in Appendix C.2.

While uniformity testing is a central problem in distribution property testing, there are many other settings where it would be interesting to develop replicable algorithms, such as closeness and independence testing [18].

---

[3] $\tilde{O}$ hides polylogarithmic factors in $n, \rho$

### 1.4 Technical Overview

$\sqrt{n}\rho^{-2}$ **Barriers for $\ell_2$ based Statistics.** Most of the well known non-replicable uniformity testers compute an unbiased estimator for the $\ell_2$ norm of the the unknown distribution $\mathbf{p}$, i.e., $\|\mathbf{p}\|_2^2 = \sum_{i=1} \mathbf{p}_i^2$, from the number of occurrences $X_i$ of each element $i$ among the observed samples. These include the testers from [28, 17], which are based on counting pair-wise collisions, i.e., $\frac{1}{2}\sum_{i=1}^n X_i(X_i - 1)$, and the ones from [14, 1, 47, 20], which are based on variants of the $\chi^2$ statistic, i.e., $\sum_{i=1}^n \left((X_i - m/n)^2 - X_i\right)/(m/n)$. As one can see, these estimators all rely heavily on computing the quantities $X_i^2$, which could have large variances even when there is a single heavy element. This poses serious challenges on designing replicable testers based on these statistics [4]. In the following paragraphs we discuss the challenge for collision based testers in more detail. A similar barrier exists for $\chi^2$ based statistics, which we defer to Appendix C.1.

Given an unknown distribution $\mathbf{p}$, the expected number of pairs of collision will be exactly $\sum_{i=1}^n \binom{m}{2}\mathbf{p}_i^2$. If $\mathbf{p}$ is uniform, the expected value of the test statistic will be about $\frac{m^2}{2n}$. If $\mathbf{p}$ is $\varepsilon$-far from being uniform, the expected value will be at least $\frac{m^2(1+\varepsilon^2)}{2n}$. To construct a replicable uniformity tester from the collision statistics, a natural idea is to select a random threshold $r$ between the two extrema to be the decision threshold. Consequently, the tester fails to replicate if and only if the random threshold falls between the realized values of the test statistics in two different runs. Conditioned on that the test statistics computed in two different runs deviate by $\Delta$, the above events happens with probability exactly $\Delta \frac{2n}{m^2\varepsilon^2}$. Hence, for the tester to be $\rho$-replicable, the test statistics will need to deviate by no more than $\Delta = O\left(m^2\varepsilon^2\rho/n\right)$ with constant probability.

To focus on the dependency on $\rho$, we let $\varepsilon$ be a small constant. We now construct a hard instance that makes collision-based statistics violate the above concentration requirement unless $m \gg \sqrt{n}\rho^{-2}$. Consider a distribution with a single heavy element with probability mass $p \gg 1/\sqrt{n}$. Let $X \sim \text{Binom}(m, p)$ denote the number of occurrences of this heavy element. It is not hard to see that this element contributes to the total collision counts by $\binom{X}{2}$, which deviates by $\Omega\left(mp\sqrt{mp}\right) = \Omega(m^{3/2}/n^{3/4})$ from its mean with constant probability. Consequently, over two runs of the algorithm, with constant probability the numbers of collisions may differ by $\Omega(m^{3/2}/n^{3/4})$. Therefore, the tester will fail to be $\rho$-replicable unless $m^{3/2}/n^{3/4} \ll \rho m^2/n$ or equivalently $m \gg \frac{\sqrt{n}}{\rho^2}$.

**Total Variation Distance Statistic** To make the test statistics less sensitive to the counts of heavy elements, we compute the *total variation statistic*, which has been used in [16] to achieve optimal uniformity testing in the high probability regime. In particular, the test statistics measures the TV distance between the empirical distribution over samples and the uniform distribution:

$$S = \frac{1}{2}\sum_{i=1}^n |X_i/m - 1/n|,$$

where $X_i$ is the number of occurrences of the $i$-th element. Unlike collision-based statistics, note that the TV statistics depends only *linearly* on each $X_i$. For a heavy element $X_i$, the contribution to the empirical total variation distance is (up to normalization) at most $X_i$ with variance $\text{Var}(X_i) = \Theta\left(mp\right) = \Theta\left(m/n^{1/2}\right)$ opposed to $\text{Var}(X_i^2) = \Theta\left(m^3/n^{3/2}\right)$. Intuitively, this allows us to obtain tighter concentration bounds on the test statistic $S$, thereby improving the final sample complexity.

First, we observe that when the distribution is $\varepsilon$-far from uniform, [16] shows that the expected value of $S$ exceeds the expected value of $S$ under the uniform distribution by at least some function $f(m, n, \varepsilon)$ (see Equation (2) for the full expression). To establish a replicable tester in the super-linear case ($m = \Theta(\sqrt{n}\varepsilon^{-2}\rho^{-1}) \geq n$), we use McDiarmid's inequality to directly argue that the TV test statistic deviates by at most $\rho f(m, n, \varepsilon)$ with high probability. Hence, if we use a random threshold that lies within the gap interval, the threshold will be $\rho f(m, n, \varepsilon)$ close to the expected value of the test statistic with probability at most $O(\rho)$, ensuring replicability of the final result.

---

[4]Interestingly, the large variance caused by heavy elements is usually not an issue in the non-replicable setting where the distribution is promised to be either uniform or far from uniform. This is because heavy elements could only exist in the latter case. In that case, the expected value of the test statistic must also be large, thus adding more slackness to the concentration requirement of the test statistics. Unfortunately, to obtain replicability, we need to deal with distributions that have heavy elements but are still relatively close to the uniform distribution.

The key challenge lies in obtaining the desired concentration in the sublinear regime, i.e. $m \leq n$. In this regime, the expectation gap is given by $f(m,n,\varepsilon) = \frac{\varepsilon^2 m^2}{n^2}$. Unfortunately, the presence of heavy elements, i.e., element with mass $\mathbf{p}_i \geq 1/m$, can still make the test statistic have high variance. As a result, it becomes challenging to obtain the desired concentration of $\rho f(m,n,\varepsilon)$ on the test statistic. However, in the presence of very heavy elements (e.g., $\mathbf{p}_i \geq m/n$) which causes the variance of the test statistic to be too large, we observe that these elements cause the expectation of the test statistic to increase sufficiently beyond the expectation gap so that the tester rejects consistently, even though the distribution may not be $\varepsilon$-far from uniform. To formalize the intuition, we show that whenever the variance of the test statistic is large, so is its expected value, thereby ensuring consistent rejection. In particular, in Lemma 3.3 we show that, whenever $\sqrt{\mathrm{Var}(S)} \gg \rho f(m,n,\varepsilon) = \frac{\rho \varepsilon^2 m^2}{n^2}$, it holds that

$$\mathbb{E}\left[S\right] - \sqrt{\mathrm{Var}(S)} \geq \mu(U_n) + \varepsilon^2 m^2 / n^2 \tag{1}$$

where $\mu(U_n)$ is the expected value of $S$ under the uniform distribution $U_n$. At a high level, Equation (1) is shown by rewriting $S$ with indicator variables representing whether some sample collides with another, and we use correlation inequalities to show that these collision indicators are "almost" independent of each other (see proof sketch of Lemma 3.3 for more detail). Combining Equation (1) with Chebyshev's inequality, we then have that $S \gg \mu(U_n) + \varepsilon^2 m^2 / n^2$ with high constant probability (which can be easily boosted to $1 - \rho$ with the standard "median trick"). As we choose our random threshold from the interval $[\mu(U_n), \mu(U_n) + \varepsilon^2 m^2 / n^2]$, it follows that the tester must consistently reject. Otherwise, we have $\sqrt{\mathrm{Var}(S)} \leq \rho f(m,n,\varepsilon)$. Applying Chebyshev's inequality gives that $\Pr\left(|S - \mathbb{E}\left[S\right]| > \rho f(m,n,\varepsilon)\right) \ll 1$. The rest of the argument is similar to the super-linear case.

**Sample complexity Lower Bound**  At a high level, we present a family of distributions $\{\mathbf{p}(\xi)\}_\xi$ parametrized by some parameter $\xi \in [0, \varepsilon]$ satisfying the following: no symmetric tester that takes fewer than $\tilde{\Omega}\left(\sqrt{n}\varepsilon^{-2}\rho^{-1}\right)$ many samples can be replicable with high probability against a random distribution from the family. Since we have fixed the testing instance distribution, using a minimax style argument similar to [30], we can assume that the testing algorithm is deterministic [5]. The family of distributions is constructed to satisfy the following properties: (i) $\mathbf{p}(0)$ is the uniform distribution and $\mathbf{p}(\varepsilon)$ is $\varepsilon$-far from uniform (ii) for any two distributions $\mathbf{p}(\xi), \mathbf{p}(\xi + \delta)$, where $\delta < \varepsilon\rho$, within the family, no symmetric tester taking fewer than $\tilde{\Omega}\left(\sqrt{n}\varepsilon^{-2}\rho^{-1}\right)$ many samples can reliably distinguish them. By (i) and the correctness guarantee of the algorithm, the acceptance probabilities of the tester should be near 1 under $\mathbf{p}(0)$ and near 0 under $\mathbf{p}(\varepsilon)$. This implies that there exists some $\xi^*$ within the range such that the acceptance probability is $1/2$ under $\mathbf{p}(\xi^*)$.[6] By property (ii), $\mathbf{p}(\xi^*)$ cannot be distinguished from $\mathbf{p}(\xi^* \pm O(\varepsilon\rho))$ by the tester, which immediately implies that the acceptance probability of the tester is near $1/2$ whenever $\xi = \xi^* \pm O(\varepsilon\rho)$, and therefore not replicable with constant probability. Thus, if we sample $\xi$ uniformly from $[0, \varepsilon]$, the tester will fail to replicate with probability at least $\Omega(\rho)$.

The construction of $\mathbf{p}(\xi)$ is natural and simple: half of the elements will have mass $(1 + \xi)/n$ and the other half will have mass $(1 - \xi)/n$. Property (i) follows immediately. The formal proof of Property (ii) is technical, but the high level intuition is straightforward. If we assume the underlying tester is symmetric, the most informative information is essentially the number of elements that have frequencies exactly 2 among the samples (as elements having frequencies more than 2 are rarely seen and the numbers of elements having frequencies 0 or 1 are about the same in the two cases.) If the tester takes $m$ samples, it observes about

$$m^2 \left((1 + \xi)^2 + (1 - \xi)^2\right)/n = 2m^2(1 + \xi^2)/n$$

many frequency-2 elements under $\mathbf{p}(\xi)$ in expectation. On the other hand, the standard deviation of the number of such elements is about $\sqrt{m^2/n} = m/\sqrt{n}$. Hence, for the tester to successfully distinguish the two distributions, $m$ needs to be sufficiently large such that

$$\frac{m}{\sqrt{n}} \ll \frac{m^2}{n}\left((1 + (\xi + \varepsilon\rho)^2) - (1 + \xi^2)\right) \approx \frac{m^2(\xi + \varepsilon\rho)\varepsilon\rho}{n},$$

---

[5]More precisely, we pick a "good random string" such that the induced deterministic tester is correct and replicable at the same time against our fixed testing instance distribution with high probability.

[6]An omitted technical detail is that our construction ensures that the acceptance probability is a continuous function of $\xi$.

which yields $m \gg \frac{\sqrt{n}}{(\xi+\varepsilon\rho)\varepsilon\rho} > \Omega\left(\frac{\sqrt{n}}{\varepsilon^2\rho}\right)$.

To formalize the above intuition, we will use an information theoretic argument [7]. Note that for such an argument to work, one often need to randomize the order of heavy and light elements [18, 16]. Otherwise, a tester could simply group the elements whose mass is above $1/n$ under $\mathbf{p}(\xi)$ into one giant bucket and reduce the problem into learning the bias of a single coin, which requires much fewer samples. To achieve this randomization, we consider the *Local Swap Family* (see Definition 4.2), where we pair the elements with mass $(1+\xi)/n$ with those with mass $(1-\xi)/n$ and randomly swap their orders. Since pairs are ordered randomly, no algorithm can hope to identify heavy/light elements without taking a significant number of samples. Thus, this creates distribution families containing $\mathbf{p}(\xi)$ and $\mathbf{p}(\xi+\delta)$ that are information theoretically hard to distinguish even for asymmetric testers. Consequently, this shows the existence of some permutation $\pi_\xi$ of $[n]$ such that the permuted distributions $\pi_\xi \cdot \mathbf{p}(\xi)$ and $\pi_\xi \cdot \mathbf{p}(\xi+\delta)$ are hard to distinguish. However, recall that in the replicability argument we need to first fix some $\xi^*$ such that $\mathbf{p}(\xi^*)=1/2$. Thus, we have to prove specifically that $\mathbf{p}(\xi)$ and $\mathbf{p}(\xi+\delta)$ themselves are hard to distinguish. Fortunately, for symmetric testers, the acceptance probabilities of $\pi_\xi \cdot \mathbf{p}(\xi)$ and $\pi_\xi \cdot \mathbf{p}(\xi+\delta)$ must be identical to those of $\mathbf{p}(\xi)$ and $\mathbf{p}(\xi+\delta)$ respectively. Consequently, no symmetric tester can easily distinguish $\mathbf{p}(\xi)$ and $\mathbf{p}(\xi+\delta)$.

## 2 Preliminaries

For any positive integer $n$, let $[n] = \{1, 2, \ldots, n\}$. We typically use $n$ to denote the domain size, $\mathbf{p}$ to denote a distribution over $[n]$ and $m$ to denote sample complexity. Given a distribution $\mathbf{p}$ over $[n]$, let $\mathbf{p}_x$ denote the probability of $x$ in $\mathbf{p}$. For a subset $S \subset [n]$, $\mathbf{p}_S = \sum_{x \in S} \mathbf{p}_x$. For distributions $\mathbf{p}, \mathbf{q}$ over $[n]$, the total variation distance is $d_{TV}(\mathbf{p}, \mathbf{q}) = \frac{1}{2}\sum_{x \in [n]} |\mathbf{p}_x - \mathbf{q}_x| = \max_{S \subseteq [n]} \mathbf{p}_S - \mathbf{q}_S$. We also recall the definition of mutual information. Let $X, Y$ be random variables over domain $\mathcal{X}$. The *mutual information* of $X, Y$ is $I(X : Y) = \sum_{x,y \in \mathcal{X}} \Pr((X,Y)=(x,y)) \log \frac{\Pr((X,Y)=(x,y))}{\Pr(X=x)\Pr(Y=y)}$. We use $a \gg b$ (resp. $a \ll b$) to denote that $a$ is a large (resp. small) constant multiple of $b$.

## 3 Replicable Uniformity Testing Algorithm

**Algorithm Overview** At a high level, our algorithm computes the TV-distance statistic and compares it with a random threshold. Correctness of the algorithm largely follows from the analysis of the test statistics from [16]. To show replicability of the algorithm, we need a better understanding of the concentration properties of the test statistic when the unknown distribution is neither uniform nor $\varepsilon$-far from being uniform. In particular, we show that when the variance of the test statistic is too large, even if the input distribution itself is not $\varepsilon$-far from uniform, the test statistic is with high probability larger than any random threshold that may be chosen, leading the algorithm to replicably reject in this case. On the other hand, when the variation is sufficiently small, we have sufficiently strong concentration in the test statistic, so that a randomly chosen threshold does not land between empirical test statistics computed from independent samples.

*Proof of Theorem 1.3.* Our starting point is the "expectation gap" of the test statistic shown in [16].

**Lemma 3.1** (Lemma 4 of [16]). *Let $\mathbf{p}$ be a distribution on $[n]$ such that $\xi = d_{TV}(\mathbf{p}, U_n)$. For any distribution $\mathbf{p}$, let $\mu(\mathbf{p})$ denote the expectation of the test statistic $S = \frac{1}{2}\sum_{i=1}^{n}\left|\frac{X_i}{m} - \frac{1}{n}\right|$ given $m$ samples drawn from $\mathbf{p}$. For all $m \geq 6, n \geq 2$, there is a constant $C$ such that*

$$\mu(\mathbf{p}) - \mu(U_n) \geq R := C \cdot \begin{cases} \xi^2 \frac{m^2}{n^2} & m \leq n \\ \xi^2 \sqrt{\frac{m}{n}} & n < m \leq \frac{n}{\xi^2} \\ \xi & \frac{n}{\xi^2} \leq m \end{cases} . \tag{2}$$

We require the following structural lemmas, whose formal proofs can be found in Appendix A, on the test statistic $S_{\text{median}}$. These lemmas show that the test statistic $S$ (and therefore $S_{\text{median}}$) concentrates

---

[7]The use of information theory in showing lower bounds for replicability has also appeared in the manuscript [29]. But our argument and construction are significantly more involved and exploit symmetries in the underlying algorithm.

---

**Algorithm 1** RUNIFORMITYTESTER$(\mathbf{p}, \varepsilon, \rho)$

---

**Input**      : Sample access to distribution $\mathbf{p}$ on domain $[n]$
**Parameters**: $\varepsilon$ tolerance, $\rho$ replicability
**Output**     : ACCEPT if $\mathbf{p} \sim U_n$ is uniform, REJECT if $d_{TV}(\mathbf{p}, U_n) \geq \varepsilon$

1  $m \leftarrow \Theta\left(\frac{\sqrt{n}}{\rho \varepsilon^2}\sqrt{\log\frac{n}{\rho}} + \frac{1}{\rho^2 \varepsilon^2}\right), m_0 \leftarrow \Theta\left(\log 1/\rho\right).$
2  **for** $1 \leq j \leq m_0$ **do**
3  $\quad$ $D_j$ gets $m$ samples from $\mathbf{p}$
4  $\quad$ $S_j \leftarrow \frac{1}{2}\sum_{i=1}^{n}\left|\frac{X_i}{m} - \frac{1}{n}\right|$ where $X_i$ is the occurrences of $i \in [n]$ in $D_j$
5  $S_{\text{median}} \leftarrow$ median of $\{S_j\}$
6  $\mu(U_n)$ is expectation of $\frac{1}{2}\sum_{i=1}^{n}\left|\frac{X_i}{m} - \frac{1}{n}\right|$ under uniform distribution.
7  Set threshold $r \leftarrow \mu(U_n) + r_0 \cdot R$ ($R$ given by Lemma 3.1) where $r_0 \leftarrow \text{Unif}\left(\frac{1}{4}, \frac{3}{4}\right)$.
8  **return** ACCEPT if $S_{\text{median}} < r$. Reject otherwise.

---

around its expectation $\mu(\mathbf{p}) = \mathbb{E}_{\mathbf{p}}[S]$ in the sublinear ($m < n$) and superlinear ($m \geq n$) cases respectively. For the superlinear case, we bound the sensitivity of $S$ with respect to the input sample set $T$ and apply McDiarmid's inequality to obtain the desired concentration result.

**Lemma 3.2** (Superlinear Concentration). *Assume that we are in the superlinear regime (i.e., $m \geq n$). Denote by $\mu(\mathbf{p})$ the expectation of the test statistic $S$ under the distribution $\mathbf{p}$. If $n \leq m \leq \frac{n}{\varepsilon^2}$, then* $\Pr\left(|S_{\text{median}} - \mu(\mathbf{p})| \geq \rho\frac{C}{16}\varepsilon^2\sqrt{\frac{m}{n}}\right) < \frac{\rho}{4}$. *If $\frac{n}{\varepsilon^2} \leq m$, then* $\Pr\left(|S_{\text{median}} - \mu(\mathbf{p})| \geq \rho\frac{C}{16}\varepsilon\right) < \frac{\rho}{4}$.

For the sublinear case, we provide a proof sketch below, deferring the details to Appendix A.

**Lemma 3.3** (Sublinear Concentration). *Suppose $m \leq n$. If $\text{Var}(S) \geq (C/64)\rho^2\varepsilon^4 m^4 n^{-4}$, then $\mathbb{E}[S] - \sqrt{\text{Var}(S)} > \mu(U_n) + C\varepsilon^2 m^2 n^{-2}$ where $\mu(U_n)$ is the expectation of $S$ under the uniform distribution and $C$ is given by Lemma 3.1. As a consequence, with probability at least $1 - \rho/4$, we have $S_{\text{median}} > \mu(U_n) + C\varepsilon^2 m^2 n^{-2}$.*

*On the other hand, if $\text{Var}(S) \leq (C/64)\rho^2\varepsilon^4 m^4 n^{-4}$, then it holds that $\Pr\left(|S_{\text{median}} - \mu(\mathbf{p})| \geq (C/16)\rho\varepsilon^2 m^2 n^{-2}\right) < \rho/4$. Furthermore, for the uniform distribution $U_n$, $\text{Var}(S) \leq (C/64)\rho^2\varepsilon^4 m^4 n^{-4}$.*

*Proof Sketch.* In the proof sketch, for simplicity, we assume that $\varepsilon$ is some small constant and ignore its dependency. When $\text{Var}(S)$ is small, we simply apply Chebyshev's inequality to obtain the desired concentration of $S$. The concentration of $S_{\text{median}}$ then follows from the standard median trick.

In the rest of the sketch we focus on the case $\text{Var}(S) \geq (C/64)\rho^2\varepsilon^4 m^4 n^{-4}$. In this case, the key is to show that
$$\mathbb{E}[S] - \sqrt{\text{Var}(S)} > \mu(U_n) + C\varepsilon^2 m^2 n^{-2} \tag{3}$$
as the claim $S_{\text{median}} > \mu(U_n) + C\varepsilon^2 m^2 n^{-2}$ with high probability follows almost immediately by an application of Chebyshev's inequality (and the standard analysis for the median trick).

Towards Equation (3), define $X_i$ as the number of occurrences of element $i \in [n]$. We begin with an observation from [16] stating that the test statistic $S = Z/n$ where $Z = |\{i \text{ s.t. } X_i = 0\}|$ denotes the number of "empty" buckets. Furthermore, we can write $Z = n - m + \sum_{i=1}^{m} Y_i$ where $Y_i$ indicates whether the $i$-th sample collides with a previous sample $j < i$. Then, $\text{Var}(S) = \text{Var}(Z)/n^2 = \text{Var}(\sum Y_i)/n^2$. To bound the variance, we argue that the indicators $Y_i$ are "almost" negatively correlated using correlation inequalities (specifically Kleitman's Theorem Lemma A.8), so that (roughly) $\text{Var}(\sum Y_i) \ll \sum_i \text{Var}(Y_i) \ll \sum \mathbb{E}[Y_i]$ (see Lemma A.5 for the accurate statement). Observe that under $U_n$, we have $\sum \mathbb{E}[Y_i] \leq m^2 n^{-1}$ so that $\mathbb{E}[S] - \mu(U_n) \geq \mathbb{E}[\sum Y_i]/n - m^2/n^2$. It follows that

$$\mathbb{E}[S] - \sqrt{\text{Var}(S)} - \mu(U_n) \geq \left(\mathbb{E}\left[\sum Y_i\right] - \sqrt{\mathbb{E}\left[\sum Y_i\right]}\right)/n - (m^2/n^2).$$

By our assumption, $\mathbb{E}[\sum Y_i] \geq n^2\text{Var}(S) \gg \rho^2\varepsilon^4 m^4 n^{-2} \gg 1$ so that it suffices to show $\mathbb{E}[\sum Y_i] \gg (C+1)m^2/n$. Since $\text{Var}(S) \geq (C/64)\rho^2\varepsilon^4 m^4 n^{-4}$ which is further bounded from below by $m^2/n^3$ whenever $m \gg \sqrt{n}\varepsilon^{-2}\rho^{-1}$ we conclude $\mathbb{E}[\sum Y_i] \geq n^2\text{Var}(S) \gg m^2/n$. $\qquad\square$

We are now ready to show Theorem 1.3 — our main algorithmic result. We begin with correctness for the uniform distribution. Suppose $m \leq n$. By Lemma 3.3, $S_{\text{median}} \leq \mu(U_n) + R/16 \leq \mu(U_n) + R/4 \leq r$ with probability at least $1 - \frac{\rho}{4}$ so the algorithm outputs ACCEPT.

Otherwise, if $n \leq m$, we have by Lemma 3.2 that with probability at least $1 - \frac{\rho}{4}$, $S_{\text{median}} \leq \mu(U_n) + R/16 \leq \mu(U_n) + R/4 \leq r$ so that the algorithm outputs ACCEPT.

On the other hand, suppose $\xi = d_{TV}(\mathbf{p}, U_n) \geq \varepsilon$. We note that the algorithm of [16] computes the test statistic $S$ using $\Theta\left(\left(\sqrt{n \log(1/\rho)} + \log(1/\rho)\right)\varepsilon^{-2}\right)$ samples and compares $S$ with some fixed threshold $R$ that is strictly larger than the random threshold $r$ of our choice. By the correctness guarantee of their algorithm, it holds that $\Pr[S > R] \geq 1 - \rho/100$, which further implies that $\Pr[S_{\text{med}} > R > r] \geq 1 - \rho/4$.

We now proceed to replicability. Consider two executions of the algorithm. If $\mathbf{p}$ is uniform or $\xi = d_{TV}(\mathbf{p}, U_n) \geq \varepsilon$, then following a union bound on the correctness condition, two executions of the algorithm output different values with probability at most $\rho/2$. Then, suppose $m \leq n$ and $\text{Var}(S) \geq (C/64)\rho^2\varepsilon^4 m^4 n^{-4}$. By Lemma 3.3, both samples lead the algorithm to output REJECT with probability at least $1 - \rho$ so that the algorithm is $\rho$-replicable.

Otherwise, Lemma 3.2 and Lemma 3.3 guarantees strong concentration of the test statistic $S_{\text{median}}$. In particular, with probability at least $1 - \rho/2$, we have $|S_{\text{median}} - \mu(\mathbf{p})| \leq \frac{\rho}{16}R$ over both samples, where $R$ is the expectation gap defined as in Equation (2). In particular, whenever the random threshold $r$ does not fall in the interval $(\mu(\mathbf{p}) \pm \rho R/16)$ both executions output the same result. Since $r$ is chosen uniformly at random, this occurs with probability at most $(\rho R/8)/(R/2) = \rho/4$. By a union bound, we observe that Algorithm 1 is $\rho$-replicable.

Finally, the sample complexity is immediately obtained by our values of $m \cdot m_0$. □

## 4   Lower Bound for Replicable Uniformity Testing

In this section, we outline the important lemmas used in showing the sample complexity lower bound for $\rho$-replicable symmetric uniformity testers, and give their proof sketches. The formal argument can be found in Appendix B.

Note that the lower bound $\tilde{\Omega}\left(\varepsilon^{-2}\rho^{-2}\right)$ holds even for testing whether the bias of a coin is $1/2$ or $1/2 + \varepsilon$ (see [30]). We therefore focus on the more challenging bound of $\tilde{\Omega}\left(\sqrt{n}\varepsilon^{-2}\rho^{-1}\right)$. Consider the canonical hard instance for uniformity testing where half of the elements have probability mass $(1 + \xi)/n$ and the other half have probability mass $(1 - \xi)/n$:

$$\mathbf{p}(\xi)_i = \left\{ \begin{array}{ll} \frac{1+\xi}{n} & \text{if } i \mod 2 = 0, \\ \frac{1-\xi}{n} & \text{otherwise.} \end{array} \right. \tag{4}$$

Our hard instance for replicable uniformity test is as follows: we choose $\xi$ from the interval $[0, \varepsilon]$ uniformly at random, and let the tester observe samples from $\mathbf{p}(\xi)$.

Fix some uniformity tester $\mathcal{A}_m$ that takes $m$ samples. We will argue that if $\mathcal{A}_m$ is $\rho$-replicable and correct with probability at least $0.99$, then we must have $m = \tilde{\Omega}(\sqrt{n}\varepsilon^{-2}\rho^{-1})$.

At a high level, we follow the framework of [30]. First, we fix some good random string $r$ such that the induced deterministic algorithm $\mathcal{A}_m(;r)$ is replicable with probability at least $1 - 10\rho$, and is correct on $\mathbf{p}(0)$ and $\mathbf{p}(\varepsilon)$ with probability at least $0.99$. Then, consider the function $\text{Acc}_m(\xi)$ that denotes the acceptance probability of $\mathcal{A}_m(S;r)$ when the samples $S$ are taken from $\mathbf{p}(\xi)$. Note that $\text{Acc}_m(\xi)$ must be a continuous function. Moreover, by the correctness of $\mathcal{A}_m(;r)$, it holds that $\text{Acc}_m(0) \geq 0.99$ and $\text{Acc}_m(\varepsilon) < 0.01$. Hence, there must be some value $\xi^*$ such that $\text{Acc}_m(\xi^*) = 1/2$. To show the desired lower bound on $m$, it suffices to show that $\text{Acc}_m(\xi^* + \delta)$ must not be too far from $\text{Acc}(\xi^*)$ for any $\delta \ll \varepsilon\rho$ if $m = \tilde{o}(\sqrt{n}\varepsilon^{-2}\rho^{-1})$.

**Proposition 4.1** (Lipschitz Continuity of Acceptance Probability). *Assume that $m = \tilde{o}(\sqrt{n}\varepsilon^{-2}\rho^{-1})$. Let $\mathcal{A}_m(;r)$ be a deterministic symmetric tester that takes $m$ samples, and define the acceptance probability function $\text{Acc}_m(\xi) = \Pr_{S \sim \mathbf{p}(\xi)^{\otimes m}}[\mathcal{A}(S;r) = 1]$, where $\mathbf{p}(\xi)$ is defined as in Equation (4). Let $\varepsilon_0 < \varepsilon_1 \in (0, \varepsilon)$ be such that $\varepsilon_1 - \varepsilon_0 < \varepsilon\rho$. Then it holds that $|\text{Acc}_m(\varepsilon_0) - \text{Acc}_m(\varepsilon_1)| < 0.1$.*

Given the above proposition, since we choose $\xi$ from $[0, \varepsilon]$ uniformly at random, it follows that the acceptance probability of the algorithm is around $1/2$ with probability at least $\Omega(\rho)$ if $m = \tilde{o}(\sqrt{n}\varepsilon^{-2}\rho^{-1})$, implying that the algorithm is not $O(\rho)$-replicable. The proof of Proposition 4.1 is based on an information theoretic argument based on ideas developed in [29]. We defer the formal proof of Proposition 4.1 to Appendix B.3 and give its proof outline below. Let $m, \varepsilon_0, \varepsilon_1$ be defined as in Proposition 4.1. At a high level, we construct two families of probability distributions, which we denote by $\mathcal{M}_0$ and $\mathcal{M}_1$, that satisfy the following properties: (i) $\mathcal{M}_i$ contains distributions that are identical to $\mathbf{p}(\varepsilon_i)$ up to domain element relabeling. (ii) Any tester that uses at most $m$ samples cannot effectively distinguish between a random probability distribution from $\mathcal{M}_0$ and a random one from $\mathcal{M}_1$. For the sake of contradiction, assume that there is a deterministic symmetric tester using $m$ samples such that the acceptance probabilities on $\mathbf{p}(\xi)$ and $\mathbf{p}(\xi + \delta)$ differ by at least $0.1$. Since all distributions within $M_0$ are identical to $\mathbf{p}(\xi)$ up to element relabeling, it follows that the acceptance probabilities of the symmetric tester on any of the distribution within $\mathcal{M}_0$ must be the same (and similarly for $\mathcal{M}_1$). This then further implies that the tester can successfully distinguish a random distribution from $\mathcal{M}_0$ versus one from $\mathcal{M}_1$, contradicting property (ii). Proposition 4.1 thereby follows.

It then remains to construct the two families of distributions. Recall that $\mathcal{M}_i$ contains distributions that are identical to $\mathbf{p}(\varepsilon_i)$ up to element relabeling. We will consider all distributions that can be obtained by performing "local swaps" on $\mathbf{p}(\varepsilon_i)$. In particular, we first group the elements into $n/2$ many adjacent pairs, and then randomly exchange the labels within each pair.

**Definition 4.2** (Local Swap Family). *Let $n$ be an even number, and $\mathbf{p}$ be a probability distribution on $[n]$. We define the Local Swap Family of $\mathbf{p}$ as the set of all distributions $\tilde{p}$ such that*

$$(\tilde{p}(\varepsilon_i)_j, \tilde{p}(\varepsilon_i)_{j+1}) = (p(\varepsilon_i)_j, p(\varepsilon_i)_{j+1}) \ \text{ or } \ (\tilde{p}(\varepsilon_i)_j, \tilde{p}(\varepsilon_i)_{j+1}) = (p(\varepsilon_i)_{j+1}, p(\varepsilon_i)_j)$$

*for all odd numbers $j \in [n]$.*

**Lemma 4.3** (Indistinguishable Distribution Families). *Let $m = \tilde{o}(\sqrt{n}\varepsilon^{-2}\rho^{-1})$, and $\varepsilon_0 < \varepsilon_1 \in (0, \varepsilon)$ be such that $\varepsilon_1 - \varepsilon_0 < \varepsilon\rho$. Let $\mathbf{p}(\varepsilon_0), \mathbf{p}(\varepsilon_1)$ be defined as in Equation (4), and $\mathcal{M}_0, \mathcal{M}_1$ be the Local Swap Families (see Definition 4.2) of $\mathbf{p}(\varepsilon_0), \mathbf{p}(\varepsilon_1)$ respectively. Let $S$ be $m$ samples drawn from either a random distribution from $\mathcal{M}_0$ or a random one from $\mathcal{M}_1$. Given only $S$, no algorithm can successfully distinguish between the two cases with probability more than $0.6$.*

The formal proof of Lemma 4.3 can be found in Appendix B.2. At a high level, we use an information theoretic argument. In particular, we consider a stochastic process where we have a random unbiased bit $X$ that controls whether we sample from a random distribution from $\mathcal{M}_0$ or a random distribution from $\mathcal{M}_1$. Let $T$ be the obtained sample set. We show that $T$ and $X$ has little mutual information. A simple application of the data processing inequality then allows us to conclude the proof. To simplify the computation involved in the argument, we will also apply the standard "Poissonization" trick. In particular, we assume that the algorithm draws $\mathrm{Poi}\,(m)$ many samples instead of exactly $m$ samples. The advantage of doing so is that we can now assume that the random variables counting the number of occurrences of each element are mutually independent conditioned on the probability distribution from which they are sampled. Moreover, one can show that this is without loss of generality by a standard reduction-based argument using the fact that Poisson distributions are highly concentrated. The formal statement of the mutual information bound is provided below.

**Lemma 4.4.** *Let $\mathcal{M}_0, \mathcal{M}_1$ be defined as in Lemma 4.3. Let $X$ be a random unbiased bit, $\tilde{p}$ a random probability distribution from $\mathcal{M}_X$, and $S$ be $\mathrm{Poi}\,(m)$ many samples from $\tilde{p}$. Moreover, let $M_i$ be the occurrences of element $i$ among $S$. Then it holds that*

$$I(X : M_1, \cdots, M_n) = O\left(\varepsilon^4 \rho^2 \frac{m^2}{n} \log^4(n)\right) + o(1).$$

To show Lemma 4.4, we first note that if we group the random variables into $n/2$ many adjacent pairs, the pairs $(M_i, M_{i+1})$ are conditionally independent and identical given $X$. Therefore, we can bound $I(X : M_1, \cdots, M_n)$ from above by $\frac{n}{2}\, I(X : M_1, M_2)$. To tackle $I(X : M_1, M_2)$, we break into three regimes depending on the relative sizez of $m, n, \varepsilon$: the sub-linear regime (approximately $m \ll n$), the super-linear regime (approximately $n < m < n/\varepsilon^2$), and the super-learning regime (approximately $m > n/\varepsilon^2$). The formal proofs involves writing the probability distributions of $M_i$s as Taylor expansions in $\varepsilon$. The calculations are rather technical and therefore deferred to Appendix B.1.

## Acknowledgments and Disclosure of Funding

The authors would like to thank Max Hopkins, Russell Impagliazzo, and Daniel Kane for many helpful discussions and suggestions.

Sihan Liu is supported by NSF Award CCF-1553288 (CAREER) and a Sloan Research Fellowship. Christopher Ye is supported by NSF Award AF: Medium 2212136, NSF grants 1652303, 1909046, 2112533, and HDR TRIPODS Phase II grant 2217058.

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

# A  Omitted Proofs for Replicable Uniformity Testing Upper Bound

## A.1  Concentration of the Total Variation Distance Statistic

We now prove the required lemmas for Theorem 1.3. This section is dedicated to proving the required lemmas regarding the concentration properties of $S_{\text{median}}$.

**Superlinear Case:** $m \geq n$

First, we consider the superlinear case, when $m \geq n$.

**Lemma 3.2** (Superlinear Concentration). *Assume that we are in the superlinear regime (i.e., $m \geq n$). Denote by $\mu(\mathbf{p})$ the expectation of the test statistic $S$ under the distribution $\mathbf{p}$. If $n \leq m \leq \frac{n}{\varepsilon^2}$, then $\Pr\left(|S_{\text{median}} - \mu(\mathbf{p})| \geq \rho\frac{C}{16}\varepsilon^2\sqrt{\frac{m}{n}}\right) < \frac{\rho}{4}$. If $\frac{n}{\varepsilon^2} \leq m$, then $\Pr\left(|S_{\text{median}} - \mu(\mathbf{p})| \geq \rho\frac{C}{16}\varepsilon\right) < \frac{\rho}{4}$.*

*Proof.* Recall that Algorithm 1 uses the median trick to boost the success probability. Here we focus on the test statistic $S := S_j$ computed in a single iteration $j \in [m_0]$. An essential tool in the analysis is McDiarmid's Inequality.

**Theorem A.1** (McDiarmid's Inequality). *Let $X_1, \ldots, X_m$ be independent random variables taking values in $\mathcal{X}$. Let $f : \mathcal{X}^m \mapsto \mathbb{R}$ be a function such that for all pairs of tuples $(x_1, \ldots, x_m), (x'_1, \ldots x'_m) \in \mathcal{X}^m$ such that $x_i = x'_i$ for all but one $i \in [m]$,*

$$|f(x_1, \ldots, x_m) - f(x'_1, \ldots, x'_m)| \leq B,$$

*Then,*

$$\Pr\left(|f(X_1, \ldots, X_m) - \mathbb{E}\left[f(X_1, \ldots, X_m)\right]| \geq t\right) < 2\exp\left(-\frac{2t^2}{mB^2}\right).$$

Observe that the samples $T_j$ are independent and a change in $T_j$ changes $S$ by at most $\frac{1}{m}$, since at most two values $X_i$ change by at most $\frac{1}{2m}$ each. Then, applying Theorem A.1 to $S = f(T_1, \ldots, T_m)$, we obtain

$$\Pr\left(|S - \mu(\mathbf{p})| \geq t\right) < 2\exp\left(-\frac{2m^2t^2}{m}\right) = 2\exp\left(-2mt^2\right). \tag{5}$$

When $n \leq m \leq \frac{n}{\varepsilon^2}$, we can conclude

$$\Pr\left(|S - \mu(\mathbf{p})| \geq \rho \cdot \frac{C}{16} \cdot \varepsilon^2\sqrt{\frac{m}{n}}\right) < 2\exp\left(-\frac{2C^2\rho^2\varepsilon^4m^2}{16^2n}\right) < \frac{1}{10}.$$

The first inequality follows from Equation (5) and the second holds whenever $m \geq \Theta\left(\frac{\sqrt{n}}{\rho\varepsilon^2}\right)$ for some sufficiently large constant factor. Finally, we show $S_{\text{median}}$ is concentrated with high probability, i.e.

$$\Pr\left(|S_{\text{median}} - \mu(\mathbf{p})| \geq \rho \cdot \frac{C}{16} \cdot \varepsilon^2\sqrt{\frac{m}{n}}\right) < \frac{\rho}{4}.$$

Note that the median fails to satisfy the concentration condition only if at least half of the intermediate statistics $S_j$ do not satisfy the concentration condition. By a Chernoff bound, this occurs with probability at most $\rho$ for $m_0 = \Theta\left(\log\frac{1}{\rho}\right)$ a sufficiently large constant.

Now consider the case $m \geq \frac{n}{\varepsilon^2}$. Note

$$\Pr\left(|S - \mu(\mathbf{p})| \geq \rho \cdot \frac{C}{16} \cdot \varepsilon\right) < 2\exp\left(-\frac{2C^2\rho^2\varepsilon^2m}{16^2}\right) < \frac{1}{10}$$

whenever $m \geq \Theta\left(\frac{1}{\rho^2\varepsilon^2}\right)$. Concentration of $S_{\text{median}}$ then follows from a similar argument as above. $\qquad\square$

**Sublinear Case:** $m \leq n$

We now consider the sublinear case. Again, we consider a single iteration $j$ and examine the concentration of the test statistic $S = S_j$. Following an observation from [16], we rewrite the test statistic as

$$S = \frac{1}{2} \sum_{i=1} \left| \frac{X_i}{m} - \frac{1}{n} \right| = \frac{1}{2} \sum_{i=1} \frac{X_i}{m} - \frac{1}{n} + \frac{2}{n} \cdot \mathbf{1}[X_i = 0] = \frac{1}{n} |\{i \text{ s.t. } X_i = 0\}| = \frac{Z}{n},$$

where we define the random variable $Z = |\{i \text{ s.t. } X_i = 0\}|$.

Our goal now is to bound the variance of $Z$. To do so, we further define some useful random variables.

**Preliminaries** Let $T \in [n]^m$ be the $m$ samples drawn, i.e., $T_i$ denotes the element corresponding to the $i$-th sample. We use $Y_1, \ldots, Y_m$ to denote whether the $i$-th sample collides with any samples $j < i$, i.e., $Y_i = 1$ if and only if there exists some $j < i$ such that $T_i = T_j$. We will consider decomposition of the domain into "heavy" and "light" elements.

**Definition A.2.** *Let* $\mathbf{p}$ *be a distribution on* $[n]$. *For each* $k \in [n]$, $k$ *is* **heavy** *if* $\mathbf{p}_k \geq \frac{3}{m} \log \frac{10n}{\rho}$ *and* **light** *otherwise. Let* $n_H, n_L$ *denote the number of heavy and light elements respectively.*

*Let* $b_{k,\mathbf{p}} = \mathbf{1}[\mathbf{p}_k \geq \frac{3}{m} \log \frac{10n}{\rho}]$ *indicate whether* $k$ *is a heavy element in* $\mathbf{p}$. *When the distribution is clear, we omit* $\mathbf{p}$ *and write* $b_k$.

We define $\tilde{Y}_1, \ldots, \tilde{Y}_m$ to indicate that the $i$-th sample comes from some light element and collides with some previous sample: $\tilde{Y}_i = Y_i \mathbf{1}\{b_{T_i} = 0\}$. Let $H$ denote the number of occurrences of heavy elements among the samples: $H = |i \text{ s.t. } b_{T_i} = 1|$. Finally, define $Z_H, Z_L$ to be the contributions to $Z$ from heavy and light elements respectively:

$$Z_H = |\{i \text{ s.t. } b_k = 1 \text{ and } X_i = 0\}|$$
$$Z_L = |\{i \text{ s.t. } b_k = 0 \text{ and } X_i = 0\}|.$$

We then show the following.

**Lemma A.3.** $\operatorname{Var}(Z) \leq \frac{\rho^3}{500} + 32 \log \frac{10n}{\rho} \sum_{i=1}^{m} \mathbb{E}[Y_i]$. *In particular,* $\operatorname{Var}(S) = \frac{\operatorname{Var}(Z)}{n^2} \leq \frac{\rho^3}{500n^2} + \frac{32}{n^2} \log \frac{10n}{\rho} \sum_{i=1}^{m} \mathbb{E}[Y_i]$.

*Proof of Lemma A.3.* Recall the random variables $Y_1, \ldots, Y_m$ to indicate whether the $i$-th sample collides with any samples $j < i$. Then, we obtain the following identities:

$$Z = n - m + \sum_{i=1}^{m} Y_i = Z_H + Z_L$$

$$Z_H = |\{i \text{ s.t. } X_i = 0 \text{ and } b_i = 1\}|$$

$$Z_L = |\{i \text{ s.t. } X_i = 0 \text{ and } b_i = 0\}| = n_L - (m - H) + \sum_{i=1}^{m} \tilde{Y}_i,$$

We will bound the variance of $Z_H, Z_L$ separately. First, we note that $Z_H = 0$ with high probability. In particular, the probability that an element with $b_k = 1$ does not occur in $m$ samples is at most $(1 - \mathbf{p}_x)^m < \left(\frac{\rho}{10n}\right)^3$, so that applying the union bound the probability that some heavy element does not occur is at most $\frac{\rho^3}{1000n^2}$, showing $\Pr(Z_H > 0) < \frac{\rho^3}{1000n^2}$. In particular,

$$\operatorname{Var}(Z_H) \leq \sum_{\ell=0}^{n} \Pr(Z_H = \ell)\ell^2$$

$$\leq n^2 \frac{\rho^3}{1000n^2}$$

$$\leq \frac{\rho^3}{1000}.$$

In the remaining proof, we bound the variance of $Z_L$. To bound the variance of $Z_L$, we separately bound the variance of $H$ and $\sum_{i=1}^{m} \tilde{Y}_i$, since $n_L, m$ are fixed constants. We begin with the first term, beginning with an upper bound on the expectation of $H$. We show the expected number of heavy elements is at most a constant multiple of the expected number of collisions.

**Lemma A.4.**
$$4 \sum_{i=1}^{m} \mathbb{E}\left[Y_i\right] \geq m \sum_{b_k=1} \mathbf{p}_k = \mathbb{E}\left[H\right] \geq \mathrm{Var}(H).$$

*Proof of Lemma A.4.* Note that $H$ is a binomial random variable so that its expectation is an upper bound on its variance. Since the $Y_i$ are non-negative random variables and $\mathbb{E}\left[Y_i\right]$ is increasing in $i$, it suffices to show the following lower bound:

$$4 \sum_{i=1}^{m} \mathbb{E}\left[Y_i\right] \geq 4 \sum_{i=m/2}^{m} \mathbb{E}\left[Y_i\right] \geq 2m\mathbb{E}\left[Y_{m/2}\right] \geq m \sum_{b_k=1} \mathbf{p}_k,$$

or equivalently

$$\mathbb{E}\left[Y_{m/2}\right] = \mathrm{Pr}(Y_{m/2} = 1) \geq \frac{1}{2} \sum_{b_k=1} \mathbf{p}_k.$$

Consider an element $k$ such that $b_k = 1$ or equivalently $\mathbf{p}_k \geq \frac{3}{m} \log \frac{10n}{\rho}$. Then, the probability that $k$ does not occur in the first $m/2$ samples is at most $(\rho n^{-1}/10)^{3/2} \leq \frac{\rho}{10n}$. Union bounding over at most $n$ elements, each element with $b_k = 1$ occurs in the first $\frac{m}{2}$ samples with probability at least $1 - \frac{\rho}{10} \geq \frac{9}{10}$. Conditioned on this, $\mathrm{Pr}(Y_{m/2} = 1) \geq \sum_{b_k=1} \mathbf{p}_k$ so that we conclude

$$\mathrm{Pr}(Y_{m/2} = 1) \geq \frac{9}{10} \sum_{b_k=1} \mathbf{p}_k.$$

$\square$

Now, we move on to the term $\sum_{i=1}^{m} \tilde{Y}_i$.

**Lemma A.5.**
$$\mathrm{Var}\left(\sum_{i=1}^{m} \tilde{Y}_i\right) \leq \left(1 + 3 \log \frac{10n}{\rho}\right) \sum_{i=1}^{m} \mathbb{E}\left[Y_i\right]$$

*Proof.* Using linearity of expectation, we have

$$\mathbb{E}\left[\left(\sum_{i=1}^{m} \tilde{Y}_i\right)^2\right] = \sum_{i=1}^{m} \mathbb{E}\left[\tilde{Y}_i^2\right] + \sum_{i \neq j} \mathbb{E}\left[\tilde{Y}_i \tilde{Y}_j\right].$$

For the first term, $\mathbb{E}\left[\tilde{Y}_i^2\right] = \mathbb{E}\left[\tilde{Y}_i\right]$ since $\tilde{Y}_i$ is an indicator variable. Considering the second term, we introduce the variables $Z_{i,j}$ denoting whether the $i$-th and $j$-th samples collide, i.e. $Z_{i,j} = \mathbf{1}[T_i = T_j]$. We can write

$$\mathbb{E}\left[\tilde{Y}_i \tilde{Y}_j\right] = \mathrm{Pr}(\tilde{Y}_i \tilde{Y}_j = 1) = \mathrm{Pr}(\tilde{Y}_i \tilde{Y}_j = 1, Z_{i,j} = 1) + \mathrm{Pr}(\tilde{Y}_i \tilde{Y}_j = 1, Z_{i,j} = 0).$$

We now bound the first term.

**Lemma A.6.**

$$\sum_{i \neq j} \mathrm{Pr}(\tilde{Y}_i \tilde{Y}_j Z_{i,j} = 1) \leq 3 \log \frac{10n}{\rho} \sum_{i=1}^{m} \mathbb{E}\left[\tilde{Y}_i\right] \leq 3 \log \frac{10n}{\rho} \sum_{i=1}^{m} \mathbb{E}\left[Y_i\right]$$

*Proof of Lemma A.6.* The event $\tilde{Y}_i, \tilde{Y}_j = 1, Z_{i,j} = 1$ happens when there is some element $k \in [n]$ such that

1. $b_k = 0$

2. both samples $T_i = T_j = k$

3. some samples from $T_{<i}$ falls in $k$. We denote this event as $E_{i,k}$.

We compute as follows:

$$
\begin{aligned}
\Pr(\tilde{Y}_i \tilde{Y}_j Z_{i,j} = 1) &= \sum_{b_k=0} \Pr(E_{i,k} = 1) \Pr(T_i = k) \Pr(T_j = k) \\
&= \sum_{b_k=0} \Pr(E_{i,k} = 1) \mathbf{p}_k^2 \\
&\leq \frac{3}{m} \log \frac{10n}{\rho} \sum_{b_k=0} \mathbf{p}_k \Pr(E_{i,k} = 1) \\
&= \frac{3}{m} \log \frac{10n}{\rho} \mathbb{E}\left[\tilde{Y}_i\right]
\end{aligned}
$$

where in the inequality we have used $b_k = 0$ gives an upper bound on $\mathbf{p}_k$. Then, for each $i$ we sum over $m - 1 \leq m$ indices $j \neq i$ to conclude the proof. $\qquad\square$

For the second term, we argue it can be upper bounded by $\Pr(\tilde{Y}_i = 1)\Pr(\tilde{Y}_j = 1)$ with Kleitman's Lemma. We define monotonically increasing and decreasing subsets.

**Definition A.7.** *Let $x, y \in \{0, 1, 2\}^n$. We say $x \preceq y$ is $x_i \leq y_i$ for all $i \in [n]$. A subset $S \subset \{0, 1, 2\}^n$ is monotonically increasing if $x \in S$ and $x \preceq y$ implies $y \in S$. $S$ is monotonically decreasing if $x \in S$ and $y \leq x$ implies $y \in S$.*

In fact, we require a small modification of Kleitman's Lemma.

**Lemma A.8** (Kleitman's Lemma [2])**.** *Let $Q$ be a distribution over random bit strings $\{0, 1, 2\}^n$. Let $\mathcal{A}, \mathcal{B}$ be two subsets of $\{0, 1, 2\}^n$ such that $\mathcal{A}$ is monotonically increasing and $\mathcal{B}$ is monotonically decreasing. Then, it holds*

$$
\Pr\left(\mathcal{A}(Q)\mathcal{B}(Q)\right) \leq \Pr\left(\mathcal{A}(Q)\right)\Pr\left(\mathcal{B}(Q)\right),
$$

*where $\mathcal{A}(Q), \mathcal{B}(Q)$ denotes the event that a random string draw from $Q$ lies in $\mathcal{A}, \mathcal{B}$.*

*Proof.* Kleitman's Lemma is proven for distributions over random bit strings $\{0, 1\}^n$, or equivalently subsets of $[n]$, replacing the $\preceq$ relation with $\subseteq$ inclusion. Given a distribution $Q$ over $\{0, 1, 2\}^n$, we design a distribution $Q'$ over $[2n]$ as follows. Let $v \in \{0, 1, 2\}^n$. We map $v$ to the subset $S = \bigcup_{i=1}^n S_i$ where $S_i = \emptyset$ if $v_i = 0$, $S_i = \{2i\}$ if $v_i = 1$ and $S_i = \{2i - 1, 2i\}$ if $v_i = 2$. Denote this mapping as $f$. This naturally induces a distribution $Q'$ over subsets of $[2n]$. Let $\mathcal{A}' = \{f(a) \text{ s.t. } a \in \mathcal{A}\}$ and $\mathcal{B}' = \{f(b) \text{ s.t. } b \in \mathcal{B}\}$.

Note that we can add sets with probability $0$ under $Q'$ to $\mathcal{A}'$ (resp. $\mathcal{B}'$) to ensure that they are monotonically increasing (resp. decreasing) without changing $\Pr(\mathcal{A}')$ (resp. $\Pr(\mathcal{B}')$). In particular, suppose there is a set $X \in \mathcal{A}'$ and $Y \notin \mathcal{A}'$ such that $X \subset Y$. Since $\mathcal{A}$ is a monotonically increasing subset of $\{0, 1, 2\}^n$, $Y \notin \mathcal{A}'$ implies that $Y_i = \{2i - 1\}$ for some $i \in [n]$, but this set has probaility $0$ under the distribution $Q'$, so we can safely add it to $\mathcal{A}'$. In particular, applying Kleitman's Lemma [2], we obtain

$$
\Pr\left(\mathcal{A}(Q)\mathcal{B}(Q)\right) = \Pr\left(\mathcal{A}'(Q')\mathcal{B}'(Q')\right) \leq \Pr\left(\mathcal{A}'(Q')\right)\Pr\left(\mathcal{B}'(Q')\right) = \Pr\left(\mathcal{A}(Q)\right)\Pr\left(\mathcal{B}(Q)\right).
$$

$\qquad\square$

Using Lemma A.8, we bound the collision probability of distinct buckets.

**Lemma A.9.** *For $i \neq j$, we have $\Pr(\tilde{Y}_i \tilde{Y}_j = 1, Z_{i,j} = 0) \leq \Pr(\tilde{Y}_i = 1)\Pr(\tilde{Y}_j = 1)$.*

*Proof of Lemma A.9.* Without loss of generality, we assume $i < j$. Recall $T_{<i} = (T_1, \ldots, T_{i-1})$.

The event $\tilde{Y}_i \tilde{Y}_j = 1, Z_{i,j} = 0$ happens when there exists two distinct elements $k \neq \ell \in [n]$ such that:

1. the $i$-th sample is $S_i = k$, the $j$-th sample is $S_j = \ell$

2. some samples from $S_{<i}$ fall in $k$, some samples from $S_{<j}$ fall in $\ell$. We denote the two events as $E_{i,k}$ and $E_{j,\ell}$ respectively.

Hence, we can write

$$\Pr[\tilde{Y}_i \, \tilde{Y}_j = 1, Z_{i,j} = 0] = \sum_{k \neq \ell, b_k = b_\ell = 0} \Pr[E_{i,k}, E_{j,\ell}] \, p_k \, p_\ell. \tag{6}$$

Fix a pair of $k < \ell$. Consider the string $\beta^{(k,\ell)}$ such that $\beta_i^{(k,\ell)} = 0$ if the $i$-th sample falls in the $k$-th bucket, $\beta_i^{(k,\ell)} = 1$ if the $i$-th sample fall in neither $k$-th nor $\ell$-th bucket, $\beta_i^{(k,\ell)} = 2$ if the $i$-th sample falls in the $\ell$-th bucket. Then, both events $E_{i,k}, E_{j,\ell}$ are completely determined by the string $\beta^{(k,\ell)}$. Moreover, the set of strings for which $E_{i,k}$ holds is monotonically decreasing and the set of strings for which $E_{j,\ell}$ holds is monotonically increasing. As a result, from Lemma A.8 we have

$$\Pr(E_{i,k} E_{j,\ell}) \leq \Pr(E_{i,k}) \Pr(E_{j,\ell}).$$

Substituting this into Equation (6) then gives us

$$\Pr(\tilde{Y}_i \tilde{Y}_j = 1, Z_{i,j=0}) \leq \sum_{k \neq \ell} \Pr(E_{j,\ell}) \Pr(E_{i,k}) p_k \, p_\ell$$

$$\leq \left( \sum_k \Pr(E_{i,k}) p_k \right) \left( \sum_\ell \Pr(E_{j,\ell}) p_\ell \right)$$

$$\leq \Pr(\tilde{Y}_i = 1) \Pr(\tilde{Y}_j = 1).$$

$\square$

To bound the total contribution to the variance of the second term, we sum and obtain

$$\sum_{i \neq j} \Pr(\tilde{Y}_i \tilde{Y}_j = 1, Z_{i,j=0}) \leq \sum_{i \neq j} \mathbb{E}\left[ \tilde{Y}_i \right] \mathbb{E}\left[ \tilde{Y}_j \right].$$

Note that $\sum_{i \neq j} \mathbb{E}\left[ \tilde{Y}_i \right] \mathbb{E}\left[ \tilde{Y}_j \right]$ is at most $\mathbb{E}\left[ \sum \tilde{Y}_i \right]^2$, so that we bound the variance as

$$\mathrm{Var}\left( \sum \tilde{Y}_i \right) = \mathbb{E}\left[ \left( \sum \tilde{Y}_i \right)^2 \right] - \mathbb{E}\left[ \sum \tilde{Y}_i \right]^2$$

$$\leq \left( 1 + 3 \log \frac{10n}{\rho} \right) \sum \mathbb{E}\left[ Y_i \right] + \sum_{i \neq j} \mathbb{E}\left[ \tilde{Y}_i \right] \mathbb{E}\left[ \tilde{Y}_j \right] - \mathbb{E}\left[ \sum \tilde{Y}_i \right]^2$$

$$\leq \left( 1 + 3 \log \frac{10n}{\rho} \right) \sum \mathbb{E}\left[ Y_i \right].$$

$\square$

We thus obtain the following bound on the variance of $Z$, proving Lemma A.3. In particular,

$$\mathrm{Var}(Z) \leq 2 \left( \mathrm{Var}(Z_H) + \mathrm{Var}(Z_L) \right)$$

$$\leq \frac{\rho^3}{500} + 4 \left( \mathrm{Var}(H) + \mathrm{Var}\left( \sum_{i=1}^m \tilde{Y}_i \right) \right)$$

$$\leq \frac{\rho^3}{500} + 20 \sum_{i=1}^m \mathbb{E}\left[ Y_i \right] + 12 \log \frac{10n}{\rho} \sum_{i=1}^m \mathbb{E}\left[ Y_i \right]$$

$$\leq \frac{\rho^3}{500} + 32 \log \frac{10n}{\rho} \sum_{i=1}^m \mathbb{E}\left[ Y_i \right].$$

$\square$

We are now ready to prove the required lemma.

**Lemma 3.3** (Sublinear Concentration). *Suppose $m \leq n$. If $\mathrm{Var}(S) \geq (C/64)\rho^2\varepsilon^4 m^4 n^{-4}$, then $\mathbb{E}[S] - \sqrt{\mathrm{Var}(S)} > \mu(U_n) + C\varepsilon^2 m^2 n^{-2}$ where $\mu(U_n)$ is the expectation of $S$ under the uniform distribution and $C$ is given by Lemma 3.1. As a consequence, with probability at least $1 - \rho/4$, we have $S_{\mathrm{median}} > \mu(U_n) + C\varepsilon^2 m^2 n^{-2}$.*

*On the other hand, if $\mathrm{Var}(S) \leq (C/64)\rho^2\varepsilon^4 m^4 n^{-4}$, then it holds that $\Pr\left(|S_{\mathrm{median}} - \mu(\mathbf{p})| \geq (C/16)\rho\varepsilon^2 m^2 n^{-2}\right) < \rho/4$. Furthermore, for the uniform distribution $U_n$, $\mathrm{Var}(S) \leq (C/64)\rho^2\varepsilon^4 m^4 n^{-4}$.*

*Proof.* Our proof considers two sub-cases. In particular, we argue that when the variance of the test statistic $S$ is large, the algorithm replicably outputs REJECT. On the other hand, when variance is small, the test statistic $S$ is tightly concentrated.

**High Variance —** $\mathrm{Var}(S) \geq (C/64)\rho^2\varepsilon^4 m^4 n^{-4}$

We argue that when the variance of the test statistic is high, the test statistic must also be large with high probability so that the algorithm replicably outputs REJECT.

First, let us consider the expectation of the test statistic if the input distribution is uniform.

**Lemma A.10.** *Suppose $m \leq n$. Let $\mu(U_n)$ be the expectation of $S$ given a sample from $U_n$. Then*

$$n \cdot \mu(U_n) \leq n - m + \frac{m^2}{n}.$$

*Proof of Lemma A.10.* For each $i$, note that there are at most $m$ distinct elements sampled before the $i$-th sample. In particular, $\mathbb{E}[Y_i] = \Pr(Y_i = 1) \leq \frac{m}{n}$ for all $i$. Then, since in the sub-linear regime $m \leq n$ we have the identities

$$S = \frac{Z}{n}$$

$$Z = n - m + \sum_{i=1}^{m} Y_i.$$

We can write the expectation of $S$ as

$$\mu(U_n) = \mathbb{E}[S] = \frac{\mathbb{E}[Z]}{n} = \frac{n - m + \sum_{i=1}^{m}\mathbb{E}[Y_i]}{n}$$

so that

$$n \cdot \mu(U_n) = n - m + \sum_{i=1}^{m}\mathbb{E}[Y_i] \leq n - m + \frac{m^2}{n}.$$

$\square$

Now, we show that when the variance of $S$ is large, with probability at least $\frac{9}{10}$, $S \geq \mu(U_n) + C\varepsilon^2\frac{m^2}{n^2}$ where $C$ is the constant given by Lemma 3.1. Since $S = \frac{Z}{n}$, this holds if and only if

$$Z \geq n \cdot \mu(U_n) + C\varepsilon^2\frac{m^2}{n}. \tag{7}$$

The expectation of $Z$ is

$$\mathbb{E}[Z] = n - m + \sum_{i=1}^{m}\mathbb{E}[Y_i].$$

From Lemma A.3, we have $\mathrm{Var}(Z) = O\left(\rho^3 + \log(n/\rho)\sum_{i=1}^{m}\mathbb{E}[Y_i]\right)$. By Chebyshev's inequality, we have that

$$\Pr\left(Z < \mathbb{E}[Z] - \sqrt{10\mathrm{Var}(Z)}\right) < \frac{1}{10}.$$

Therefore, it suffices to show $\mathbb{E}[Z] - \sqrt{10\mathrm{Var}(Z)} \geq n \cdot \mu(U_n) + C\varepsilon^2 \frac{m^2}{n^2}$. If this holds, then $S$ is beyond the random threshold and the algorithm outputs REJECT. We rearrange the desired inequality as follows:

$$\mathbb{E}[Z] - \sqrt{10\mathrm{Var}(Z)} \geq n \cdot \mu(U_n) + C\varepsilon^2 \frac{m^2}{n^2}$$

$$\mathbb{E}[Z] - n \cdot \mu(U_n) - \sqrt{10\mathrm{Var}(Z)} \geq C\varepsilon^2 \frac{m^2}{n^2}.$$

Plugging in $\mathbb{E}[Z] - n \cdot \mu(U_n) \geq \sum_{i=1}^{m} \mathbb{E}[Y_i] - \frac{m^2}{n}$, it suffices to show

$$\sum_{i=1}^{m} \mathbb{E}[Y_i] - \frac{m^2}{n} - \sqrt{10\mathrm{Var}(Z)} \geq C\varepsilon^2 \frac{m^2}{n},$$

which is true as long as

$$\sum_{i=1}^{m} \mathbb{E}[Y_i] - \sqrt{10\mathrm{Var}(Z)} \geq 2C\frac{m^2}{n}.$$

Thus, applying Lemma A.3, it remains to show

$$\sum_{i=1}^{m} \mathbb{E}[Y_i] - \sqrt{\frac{\rho^3}{50} + 320\log\frac{10n}{\rho}\sum_{i=1}^{m}\mathbb{E}[Y_i]} \geq \sum_{i=1}^{m} \mathbb{E}[Y_i] - 1 - \sqrt{320\log\frac{10n}{\rho}\sum_{i=1}^{m}\mathbb{E}[Y_i]} \geq 2C\frac{m^2}{n}.$$

since $\sqrt{a+b} \leq \sqrt{a} + \sqrt{b}$ and $\sqrt{\rho^3/50} \ll 1$. Finally, we observe that by our choice of $m$, we have $2Cm^2n^{-1} \gg 1$ so that it suffices to show

$$\sum_{i=1}^{m} \mathbb{E}[Y_i] - \sqrt{320\log\frac{10n}{\rho}\sum_{i=1}^{m}\mathbb{E}[Y_i]} \geq 3C\frac{m^2}{n}.$$

We simplify the left hand side with the following lemma.

**Lemma A.11.** *For any $x \geq 1280\log\frac{10n}{\rho}$, we have*

$$x - \sqrt{320\log\frac{10n}{\rho}x} \geq \frac{x}{2}.$$

*Proof.* Suppose $x \geq 320\log\frac{10n}{\rho}$. Then the inequality holds if and only if

$$\frac{x}{2} \geq \sqrt{320\log\frac{10n}{\rho}x}$$

$$\sqrt{x} \geq 2\sqrt{320\log\frac{10n}{\rho}}$$

$$x \geq 1280\log\frac{10n}{\rho}.$$

$\square$

We now lower bound $\sum_{i=1}^{m} \mathbb{E}[Y_i]$. First, from Lemma A.3 we have

$$\sum_{i=1}^{m} \mathbb{E}[Y_i] \geq \frac{\mathrm{Var}(Z) - \frac{\rho^3}{500}}{32\log(10n/\rho)} = \frac{\mathrm{Var}(S)n^2 - \frac{\rho^3}{500}}{32\log(10n/\rho)}.$$

By assumption on $\mathrm{Var}(S) \geq (C/64)\rho^2\varepsilon^4m^4n^{-4}$, we obtain

$$\sum_{i=1}^{m} \mathbb{E}[Y_i] \geq \frac{(C/64)\rho^2\varepsilon^4m^4n^{-2} - \rho^3/500}{32\log(10n/\rho)}.$$

Finally, since $m \gg \sqrt{n}\rho^{-1}\varepsilon^{-2}\sqrt{\log(n/\rho)}$ for some sufficiently large constant, we have

$$(C/64)\rho^2\varepsilon^4 m^4 n^{-2} - \rho^3/500 \gg \frac{\log^2(n/\rho)}{\rho^2\varepsilon^4}.$$

Then,

$$\sum_{i=1}^{m} \mathbb{E}\left[Y_i\right] \gg \frac{\log^2(n/\rho)}{\rho^2\varepsilon^4 \log(n/\rho)} \geq 1280 \log \frac{n}{\rho}.$$

Thus, we conclude by

$$\sum_{i=1}^{m} \mathbb{E}\left[Y_i\right] - \sqrt{320 \log \frac{10n}{\rho} \sum_{i=1}^{m} \mathbb{E}\left[Y_i\right]} \geq \frac{1}{2} \sum_{i=1}^{m} \mathbb{E}\left[Y_i\right]$$

$$\geq 2C\frac{m^2}{n}.$$

where the final equality follows from

$$\sum_{i=1}^{m} \mathbb{E}\left[Y_i\right] \geq \frac{(C/64)\rho^2\varepsilon^4 m^4 n^{-2} - \rho^3/500}{56\log(10n/\rho)} \gg \frac{\log^2(n/\rho)}{\rho^2\varepsilon^4 \log(n/\rho)} = \frac{\log(n/\rho)}{\rho^2\varepsilon^4} \tag{8}$$

and

$$\frac{m^2}{n} = O\left(\frac{\log(n/\rho)}{\rho^2\varepsilon^4}\right).$$

and if we choose an arbitrary large constant factor of $m$, left hand side $\sum \mathbb{E}\left[Y_i\right]$ takes this constant to the 4th power while the right hand side takes this constant to the 2nd power.

Thus, whenever the variance of $S$ is large, the test statistic $S$ lies above any random threshold with probability at least $\frac{9}{10}$. Again applying a Chernoff bound, $S_{\mathrm{median}}$ does not lie above any random threshold with probability at most $\frac{\rho}{4}$.

For the high variance case, it remains to show the inequality

$$\mathbb{E}\left[S\right] - \sqrt{\mathrm{Var}(S)} \geq \mu(U_n) + C\varepsilon^2 \frac{m^2}{n^2}.$$

From the identities $S = Z/n$ and $Z = n - m + \sum Y_i$, Lemma A.3 and Lemma A.10, we can rewrite the above equation as

$$\frac{\sum \mathbb{E}\left[Y_i\right] - m^2/n}{n} - \sqrt{\frac{\rho^3}{500n^2} + \frac{32}{n^2}\log\frac{10n}{\rho}\sum \mathbb{E}\left[Y_i\right]} \geq C\varepsilon^2\frac{m^2}{n^2}$$

and we can lower bound the left hand side as

$$\frac{\sum \mathbb{E}\left[Y_i\right] - \frac{m^2}{n}}{n} - \frac{1}{n} - \sqrt{\frac{32}{n^2}\log\frac{10n}{\rho}\sum \mathbb{E}\left[Y_i\right]} \geq \frac{\sum \mathbb{E}\left[Y_i\right] - \sqrt{32\log\frac{10n}{\rho}\sum \mathbb{E}\left[Y_i\right]}}{n} - \frac{m^2}{n^2} - \frac{1}{n}$$

$$\geq \frac{\sum \mathbb{E}\left[Y_i\right]}{2n} - \frac{m^2}{n^2} - \frac{1}{n}$$

where we obtain the first term by using $\sqrt{a+b} \leq \sqrt{a} + \sqrt{b}$ and $\rho^3/(500n^2) \leq n^{-2}$, the second term by grouping similar terms, and the final term by our assumption on $\mathrm{Var}(S)$ and therefore $\mathbb{E}\left[\sum Y_i\right]$. Therefore, it suffices to show

$$\sum \mathbb{E}\left[Y_i\right] \geq 1 + \frac{m^2}{n} + C\varepsilon^2\frac{m^2}{n}.$$

or since $m^2 n^{-1} \gg 1$,

$$\sum \mathbb{E}\left[Y_i\right] \geq 4\max(C,1)\frac{m^2}{n}$$

which follows from the lower bound on $\sum \mathbb{E}\left[Y_i\right]$ given by Equation (8).

**Low Variance** — $\mathrm{Var}(S) \le (C/64)\rho^2\varepsilon^4 m^4 n^{-4}$

We now show that whenever the number of expected collisions is low, the test statistic is well concentrated. By Chebyshev's inequality, we have for any iteration $j$ that

$$\Pr\left(|S_j - \mu(\mathbf{p})| > \frac{C}{16}\rho\varepsilon^2 \frac{m^2}{n^2}\right) = \Pr\left(|S_j - \mu(\mathbf{p})| > 4 \cdot \frac{C}{64}\rho\varepsilon^2 \frac{m^2}{n^2}\right) < \frac{1}{10}.$$

Using a Chernoff bound as previously, we obtain the same concentration result for $S_{\mathrm{median}}$ with high probability.

Finally, we show that for the uniform distribution $U_n$, we have

$$\mathrm{Var}(S) \le (C/64)\rho^2\varepsilon^4 m^4 n^{-4}.$$

We observe that for the uniform distribution, $\mathbb{E}\left[Y_i\right] \le \frac{m}{n}$ since the $i$-th sample can collide with at most $i \le m$ elements before it and therefore $\sum_{i=1}^m \mathbb{E}\left[Y_i\right] \le \frac{m^2}{n}$. Then, from Lemma A.3,

$$\mathrm{Var}(S) = \frac{\mathrm{Var}(Z)}{n^2} \le \frac{24\log(10n/\rho)}{n^2}\sum_{i=1}^m \mathbb{E}\left[Y_i\right] \le \frac{24m^2\log(10n/\rho)}{n^3}.$$

We thus require

$$\frac{24m^2\log(10n/\rho)}{n^3} \le (C/64)\rho^2\varepsilon^4 m^4 n^{-4}$$
$$\frac{1536n\log(10n/\rho)}{C\rho^2\varepsilon^4} \le m^2$$
$$\frac{\sqrt{n}}{\rho\varepsilon^2}\sqrt{\log\frac{n}{\rho}} \ll m$$

which is satisfied by our sample complexity $m$. $\qquad\square$

# B  Omitted Proofs for Replicable Uniformity Testing Lower Bound

## B.1  Proof of Lemma 4.4

We begin by bounding the mutual information of the sub-linear regime.

**Lemma B.1.** *Let $\frac{m}{n} \le 1$. Let $\delta = |\varepsilon_0 - \varepsilon_1|$ and $\varepsilon = \max(\varepsilon_0, \varepsilon_1)$. Then,*

$$\sum_{a,b\in\mathbb{N}} \frac{(\Pr(M_1 = a, M_2 = b \mid X = 0) - \Pr(M_1 = a, M_2 = b \mid X = 1))^2}{\Pr(M_1 = a, M_2 = b)} \le O\left(\frac{\varepsilon^2\delta^2 m^2}{n^2}\right). \quad (9)$$

*Proof.* First, let us expand the conditional probabilities for a fixed $a, b$. Expanding the definition of the random variables $M_1, M_2, X$ and applying the probability mass function of Poisson distributions, we arrive at the expression

$$\Pr(M_1 = a, M_2 = b \mid X = 0) = \frac{1}{2a!b!}\exp\left(-\frac{2m}{n}\right)\left(\frac{m}{n}\right)^{a+b}\left((1+\varepsilon_0)^a(1-\varepsilon_0)^b + (1-\varepsilon_0)^a(1+\varepsilon_0)^b\right),$$

$$\Pr(M_1 = a, M_2 = b \mid X = 1) = \frac{1}{2a!b!}\exp\left(-\frac{2m}{n}\right)\left(\frac{m}{n}\right)^{a+b}\left((1+\varepsilon_1)^a(1-\varepsilon_1)^b + (1-\varepsilon_1)^a(1+\varepsilon_1)^b\right)$$

Define the function

$$f_{a,b}(x) := (1+x)^a(1-x)^b + (1-x)^a(1+x)^b.$$

Then we have

$$\sum_{a,b\in\mathbb{N}} \frac{(\Pr(M_1=a,M_2=b\mid X=0) - \Pr(M_1=a,M_2=b\mid X=1))^2}{\Pr(M_1=a,M_2=b)}$$

$$= O(1) \sum_{a,b\in\mathbb{N}} \frac{1}{a!b!} \exp\left(-\frac{2m}{n}\right) \left(\frac{m}{n}\right)^{a+b} \frac{(f_{a,b}(\varepsilon_1) - f_{a,b}(\varepsilon_0))^2}{f_{a,b}(\varepsilon_1) + f_{a,b}(\varepsilon_0)}$$

$$\leq O(1) \sum_{a,b\in\mathbb{N}} \frac{1}{a!b!} \exp\left(-\frac{2m}{n}\right) \left(\frac{m}{n}\right)^{a+b} \frac{\max_{\varepsilon_0\leq x\leq\varepsilon_1}\left(\frac{\partial}{\partial x}f_{a,b}(x)\right)^2}{f_{a,b}(\varepsilon_0) + f_{a,b}(\varepsilon_1)} (\varepsilon_1 - \varepsilon_0)^2 , \qquad (10)$$

where in the last line we use the mean value theorem. The main technical step will be to bound from above the quantity $\frac{\max_{\varepsilon_0\leq x\leq\varepsilon_1}\left(\frac{\partial}{\partial x}f_{a,b}(x)\right)^2}{f_{a,b}(\varepsilon_0)+f_{a,b}(\varepsilon_1)}$. Specifically, we show the following technical claim.

**Claim B.2.** *There exists an absolute constant $C$ such that*

$$\max_{\varepsilon_0<x<\varepsilon_1} \frac{\left(\frac{\partial}{\partial x}f_{a,b}(x)\right)^2}{(f_{a,b}(\varepsilon_0) + f_{a,b}(\varepsilon_1))} = C \begin{cases} 0 & \text{if } a+b \leq 1 \\ (a^4+b^4)\varepsilon^2 & \text{if } 1 < a+b \leq \varepsilon^{-1}/2 \\ (a+b)^2 \left(\frac{1+\varepsilon}{1-\varepsilon}\right)^{a+b} & \text{if } a+b > \varepsilon^{-1}/2. \end{cases}$$

*Proof.* If $a+b\leq 1$, $f_{a,b}(x) = 2$. Thus, $\frac{d}{dx}f_{a,b}(x) = 0$, which gives the first case.

We next analyze the case $2 \leq a+b \leq \varepsilon^{-1}$. Note that $f_{a,b}(x)$ is an even function with respect to $x$, i.e., $f_{a,b}(x) = f_{a,b}(-x)$. This allows us to write $f_{a,b}(x) = \frac{1}{2}(f_{a,b}(x) + f_{a,b}(-x))$. As a result, we can conclude that $f_{a,b}(x)$ does not contain any monomial of $x$ with odd degree. Thus, we can write

$$f_{a,b}(x) = c_{a,b}^{(0)} + \sum_{d\in[a+b] \text{ is even}} c_{a,b}^{(d)}x^d$$

for some coefficients $c_{a,b}^{(d)}$ that depend on $a,b$ only. This implies that

$$\frac{\partial}{\partial x}f_{a,b}(x) = \sum_{d\in[a+b] \text{ is even}} d c_{a,b}^{(d)}x^{d-1}.$$

When $2 \leq a+b \leq \varepsilon^{-1}/2$, the coefficients $c_{a,b}^{(d)}$ is of order $\binom{a+b}{d}$. Since $x < \varepsilon$, we must have that $|d\, c_{a,b}^{(d)}\, x^{d-1}|$ decreases exponentially fast in $d$. This shows that $|\frac{\partial}{\partial x}f_{a,b}(x)|$ is dominated by the contribution from the monomial $|c_{a,b}^{(2)}x|$, which implies that

$$\max_{\varepsilon_0<x<\varepsilon_1} \left|\frac{\partial}{\partial x}f_{a,b}(x)\right| \leq O(1)\,(a+b)^2\varepsilon. \qquad (11)$$

To finish the analysis for this case, it then suffices to bound from below $f_{a,b}(\varepsilon_0) + f_{a,b}(\varepsilon_1)$. Without loss of generality, assume that $a \leq b$. Then we have

$$f_{a,b}(\varepsilon_1) \geq (1 - \varepsilon_1^2)^a \geq \Omega(1 - a\varepsilon_1^2) \geq \Omega(1), \qquad (12)$$

where the last inequality follows from $a \leq a+b < \varepsilon^{-1}/2$ and $\varepsilon_1 < \varepsilon$. Combining Equation (11) and Equation (12) then concludes the proof of the second case.

Lastly, we analyze the case $(a+b) > \varepsilon^{-1}/2$. In this case, for $\varepsilon_0 \leq x \leq \varepsilon_1$, we note that

$$\frac{\partial}{\partial x}f_{a,b}(x) = (1+x)^a(1-x)^b \left(\frac{a}{1+x} - \frac{b}{1-x}\right) + (1-x)^a(1+x)^b \left(\frac{b}{1+x} - \frac{a}{1-x}\right).$$

It then follows that

$$\left|\frac{\partial}{\partial x}f_{a,b}(x)\right| \leq O(a+b)\left((1+x)^a(1-x)^b + (1-x)^a(1+x)^b\right)$$

$$\leq O(a+b)\left((1+x)^a + (1+x)^b\right), \qquad (13)$$

where in the first inequality we use that $|u - v| \leq u + v$ for any $u, v > 0$, and the second inequality simply follows from $1 - x < 1$. We can therefore conclude that

$$
\max_{\varepsilon_0 \leq x \leq \varepsilon_1} \frac{\left(\frac{\partial}{\partial x} f_{a,b}(x)\right)^2}{f_{a,b}(\varepsilon_0) + f_{a,b}(\varepsilon_1)} \leq O(1)(a+b)^2 \frac{\left((1+\varepsilon_1)^a + (1+\varepsilon_1)^b\right)^2}{(1+\varepsilon_1)^a(1-\varepsilon_1)^b + (1+\varepsilon_1)^b(1-\varepsilon_1)^a}
$$
$$
\leq O(1)(a+b)^2 \left(\frac{(1+\varepsilon_1)^a}{(1-\varepsilon_1)^b} + \frac{(1+\varepsilon_1)^b}{(1-\varepsilon_1)^a}\right)
$$
$$
\leq O(1)(a+b)^2 \left(\frac{1+\varepsilon_1}{1-\varepsilon_1}\right)^{a+b},
$$

where in the first inequality we bound from above the numerator with the square of the right hand side of Equation (13) substituted with $x = \varepsilon_1$, and bound from below the denominator by $f_{a,b}(\varepsilon_1)$, in the second inequality we use the fact that $(u + v)^2 \leq 2u^2 + 2v^2$, and in the last inequality we use that $(1 + \varepsilon_1)^a$, $(1 - \varepsilon_1)^{-a}$ are increasing in $a$ and $(1 + \varepsilon_1)^b$, $(1 - \varepsilon_1)^{-b}$ are increasing in $b$. This concludes the proof of the third case and also Claim B.2. $\qquad\square$

It immediately follows from the claim that

$$
\sum_{a,b:a+b<2} \frac{(\Pr(M_1 = a, M_2 = b \mid X = 0) - \Pr(M_1 = a, M_2 = b \mid X = 1))^2}{\Pr(M_1 = a, M_2 = b)} = 0. \quad (14)
$$

Consider the terms with $2 \leq a + b \leq \varepsilon^{-1}/2$. From Equation (10) and Claim B.2, we have that

$$
\sum_{a,b:2\leq a+b\leq\varepsilon^{-1}/2} \frac{(\Pr(M_1 = a, M_2 = b \mid X = 0) - \Pr(M_1 = a, M_2 = b \mid X = 1))^2}{\Pr(M_1 = a, M_2 = b)}
$$
$$
\leq O(1) \sum_{a,b:2\leq a+b\leq\varepsilon^{-1}/2} \frac{1}{a!b!} \exp\left(-\frac{2m}{n}\right) \left(\frac{m}{n}\right)^{a+b} (a^4 + b^4)\delta^2\varepsilon^2
$$
$$
= O(\delta^2\varepsilon^2) \left(\sum_{a\in\mathbb{N}} \sum_{b:2\leq a+b\leq\varepsilon^{-1}/2} \frac{a^4}{a!b!} \left(\frac{m}{n}\right)^{a+b} + \sum_{b\in\mathbb{N}} \sum_{a:2\leq a+b\leq\varepsilon^{-1}/2} \frac{b^4}{a!b!} \left(\frac{m}{n}\right)^{a+b}\right). \quad (15)
$$

Define $A := \sum_{a\in N} \sum_{b:b:2\leq a+b\leq\varepsilon^{-1}/2} \frac{a^4}{a!b!} \left(\frac{m}{n}\right)^{a+b}$ and $B := \sum_{b\in N} \sum_{a:2\leq a+b\leq\varepsilon^{-1}/2} \frac{b^4}{a!b!} \left(\frac{m}{n}\right)^{a+b}$. One can see the two terms are similar to each other. So we focus on the term $A$. We have that

$$
A = \sum_{b=1}^{\varepsilon^{-1}/2-1} \frac{1}{b!} \left(\frac{m}{n}\right)^{1+b} + \sum_{a\geq 2} \frac{a^4}{a!} \left(\frac{m}{n}\right)^a \sum_{b:b:2\leq a+b\leq\varepsilon^{-1}/2} \frac{1}{b!} \left(\frac{m}{n}\right)^b
$$
$$
\leq O(1) \left(\frac{m}{n}\right)^2 + O(1) \sum_{a\geq 2} \frac{a^4}{a!} \left(\frac{m}{n}\right)^a.
$$
$$
\leq O(1) \left(\frac{m}{n}\right)^2 + O(1) \exp(m/n) \sum_{a\geq 2} \exp(-m/n) \frac{a(a-1) + a(a-1)(a-2)(a-3)}{a!} \left(\frac{m}{n}\right)^a
$$
$$
\leq O(1) \left(\frac{m}{n}\right)^2 + O(1) \exp(m/n) \mathbb{E}_{y\sim\text{Poi}(m/n)} \left[y(y-1) + y(y-1)(y-2)(y-3)\right],
$$
$$
\leq O(1) \left(\frac{m}{n}\right)^2.
$$

where in the first inequality we use the assumption $m < n$ to bound from the above the summation over $b$ by a geometric series with common ratio at most $1/2$, in the second inequality we use the fact that $a^4$ is at most $10 \ (a(a-1) + a(a-1)(a-2)(a-3))$ for all $a \geq 2$, in the third inequality we observe that the summation over $a$ can be bounded from above by the moments of some Poisson random variable with mean $m/n$, and in the last inequality we use the fact that

$\mathbb{E}_{y \sim \mathrm{Poi}(\lambda)}[y(y-1) + y(y-1)(y-2)(y-3)] = \lambda^2 + \lambda^4$. One can show the same upper bound for $B$ via an almost identical argument. Substituting the bounds for $A, B$ into Equation (15) then gives

$$\sum_{a,b:2 \leq a+b \leq \varepsilon^{-1}/2} \frac{(\Pr(M_1 = a, M_2 = b \mid X = 0) - \Pr(M_1 = a, M_2 = b \mid X = 1))^2}{\Pr(M_1 = a, M_2 = b)} \leq O(\delta^2 \varepsilon^2) \left( \frac{m}{n} \right)^2.$$

(16)

It then remains to analyze the terms where $a + b \geq \varepsilon^{-1}/2$. Again from Equation (10) and Claim B.2, we have that

$$\sum_{a,b \in \mathbb{N}} \frac{(\Pr(M_1 = a, M_2 = b \mid X = 0) - \Pr(M_1 = a, M_2 = b \mid X = 1))^2}{\Pr(M_1 = a, M_2 = b)}$$

$$\leq O(\delta^2) \sum_{a,b:a+b > \varepsilon^{-1}/2} \frac{1}{a!b!} \left( \frac{m}{n} \right)^{a+b} (a+b)^2 \left( \frac{1+\varepsilon}{1-\varepsilon} \right)^{a+b}$$

$$= O(\delta^2) \sum_{s > \varepsilon^{-1}/2} \left( \frac{m}{n} \frac{1+\varepsilon}{1-\varepsilon} \right)^s s^2 \sum_{a,b:a+b=s} \frac{1}{a!b!}$$

$$\leq O(\delta^2) \sum_{s > \varepsilon^{-1}/2} \left( \frac{m}{n} \frac{1+\varepsilon}{1-\varepsilon} \right)^s \frac{s^3}{\lfloor s/2 \rfloor!}$$

$$\leq O(\delta^2) \frac{s^3}{\lfloor s/2 \rfloor!} \left( \frac{1}{2} \frac{m}{n} \frac{1+\varepsilon}{1-\varepsilon} \right)^s \bigg|_{s=\varepsilon^{-1}/2} \leq O(\delta^2 \varepsilon^2 (m/n)^2) \qquad (17)$$

where in the second inequality we use the observation that we must have either $a \geq s/2$ or $b \geq s/2$ if $a + b = s$, in the third inequality we note that the summation over $s$ decreases exponentially and is therefore dominated by the term where $s = \varepsilon^{-1}/2$, and in the last inequality we use that $s^3/\lceil s/2 \rceil! \leq O(1)$, $(m/n)^{\varepsilon^{-1}/2} \leq (m/n)^2$, $(1+\varepsilon)/(2(1-\varepsilon)) < 0.9$, and that $0.9^{\varepsilon^{-1}/2} \ll \mathrm{poly}\,(\varepsilon)$ for sufficiently small $\varepsilon$.

Combining Equations (14), (16) and (17) then concludes the proof of Lemma B.1. $\qquad \square$

We now bound the mutual information for the super-linear regime.

**Lemma B.3.** *Let $\delta = |\varepsilon_0 - \varepsilon_1|$ and $\varepsilon = \max(\varepsilon_0, \varepsilon_1)$. Let $n < m < o\left( \frac{n}{\log^2 n \, \varepsilon^2} \right)$. Then,*

$$\sum_{a,b \in \mathbb{N}} \frac{(\Pr(M_1 = a, M_2 = b \mid X = 0) - \Pr(M_1 = a, M_2 = b \mid X = 1))^2}{\Pr(M_1 = a, M_2 = b)} \leq O\left( \frac{\varepsilon^2 \delta^2 m^2 \, \log^4 n}{n^2} \right) + o(1/n).$$

*Proof.* We note that for a Poisson random variable $y$ with mean $\lambda \geq 1$, we have $|y - \lambda| > \sqrt{\lambda} \log n$ with probability at most $o(1/n)$. We can focus on the terms where $l \in [\lambda - \sqrt{\lambda} \log n, \lambda + \sqrt{\lambda} \log n]$. For reasons that will become clear soon, we will assume that $a, b$ both lie in this range.

As in the sub-linear case, we define the function

$$f_{a,b}(x) := (1+x)^a (1-x)^b + (1-x)^a (1+x)^b.$$

Instead of computing the derivative of $f_{a,b}$ directly, we will first rewrite it slightly.

$$f_{a,b}(x) = \exp(a \log(1+x) + b \log(1-x)) + \exp(a \log(1-x) + b \log(1+x))$$

$$= 2 + a \log(1-x^2) + b \log(1-x^2)$$

$$+ \sum_{i=2}^{\infty} \frac{(a \log(1+x) + b \log(1-x))^i + (a \log(1-x) + b \log(1+x))^i}{i!},$$

where in the second equality we apply the Taylor approximation of the exponential function. We will analyze the expression terms by terms. In particular, define

$$g_y(x) = y \, \log(1-x^2), \, h_{a,b}(x) = \sum_{i=2}^{\infty} \frac{(a \log(1+x) + b \log(1-x))^i}{i!}.$$

Then we have

$$f_{a,b}(x) = 2 + g_a(x) + g_b(x) + h_{a,b}(x) + h_{a,b}(-x). \tag{18}$$

We first analyze $g_a(x)$. When $a \in [\lambda - \sqrt{\lambda} \log n, \lambda + \sqrt{\lambda} \log n]$, we have that

$$
\begin{aligned}
|g_a(\varepsilon_1) - g_a(\varepsilon_0)| &= \left| a \log \left( \frac{1 - \varepsilon_0^2}{1 - \varepsilon_1^2} \right) \right| \le a \left| 1 - \frac{1 - \varepsilon_0^2}{1 - \varepsilon_1^2} \right| \\
&= \frac{a}{1 - \varepsilon_1^2} (\varepsilon_0 + \varepsilon_1) |\varepsilon_0 - \varepsilon_1| \le O \left( \frac{m}{n} \varepsilon \delta \log n \right),
\end{aligned} \tag{19}
$$

where in the first inequality we use the fact $|\log(x)| \le x - 1$ for $x \ge 1$, in the second inequality we use our assumption that $\max(\varepsilon_0, \varepsilon_1) \le \varepsilon$ and $|\varepsilon_0 - \varepsilon_1| \le \delta$, and our assumption that $|a - m/n| < \log n \sqrt{m/n}$. Using a similar argument, one can also derive that

$$|g_b(\varepsilon_1) - g_b(\varepsilon_0)| \le O \left( \frac{m}{n} \varepsilon \delta \log n \right), \tag{20}$$

We then turn to the term $h_{a,b}(x)$. The derivative with respect to $x$ is given by

$$
\begin{aligned}
\frac{\partial}{\partial x} h_{a,b}(x) &= \left( \frac{a}{1 + x} - \frac{b}{1 - x} \right) \sum_{i=2}^{\infty} \frac{(a \log(1 + x) + b \log(1 - x))^{i-1}}{(i-1)!} \\
&= \left( \frac{a}{1 + x} - \frac{b}{1 - x} \right) \left( (1 + x)^a (1 - x)^b - 1 \right)
\end{aligned}
$$

where the second equality follows from the observation that the summation is exactly the Taylor approximation of $\exp(a \log(1 + x) + b \log(1 - x))$ without the constant term. We next proceed to bound from above $\left| \frac{\partial}{\partial x} h_{a,b}(x) \right|$. First, when $x$ is sufficiently small and $a, b$ lie in the range $\frac{m}{n} \pm \sqrt{\frac{m}{n}} \log n$, note that

$$
\begin{aligned}
\left| \frac{a}{1 + x} - \frac{b}{1 - x} \right| &= |a - b - O(x)(a + b)| \\
&\le |a - b| + (a + b)O(x) \\
&\le 2\sqrt{\frac{m}{n}} \log n + 2\frac{m}{n} x + 2\sqrt{\frac{m}{n}} \log(n) x \\
&\le O \left( \sqrt{\frac{m}{n}} \log n \right),
\end{aligned} \tag{21}
$$

where in the first line we approximate $\frac{1}{1+x}$ and $\frac{1}{1-x}$ with $1 - O(x)$ and $1 + O(x)$ respectively for sufficiently small $x$, the second line follows from the triangle inequality, in the third line we use the assumption on the ranges of $a, b$, in the last line we note that $\sqrt{\frac{m}{n}} \log n$ is the dominating term when $x < \varepsilon$ and $m < n/\varepsilon^2$. Next, under the same set of assumptions, we have that

$$
\begin{aligned}
\left( (1 + x)^a (1 - x)^b - 1 \right) &\le \left( (1 - x^2)^b (1 + x)^{|a-b|} - 1 \right) \\
&\le \left( (1 - O(bx^2))(1 + O(|a - b|)x) - 1 \right) \\
&\le O \left( \varepsilon \sqrt{\frac{m}{n}} \log n \right),
\end{aligned} \tag{22}
$$

where in the second inequality we note that since $bx^2 = O\left( \frac{m}{n} \varepsilon^2 \right) \ll 1$ and $|a - b|x = O\left( \sqrt{\frac{m}{n}} \log(n)\varepsilon \right) \ll 1$ we can approximate $(1 - x^2)^b$ and $(1 + x)^{|a-b|}$ by $(1 - O(bx^2))$ and $(1 + O(|a - b|)x)$ respectively, and in the last inequality we use the assumption on the range of $a, b$ and $x < \varepsilon$. Thus, combining Equations (21) and (22), we arrive at the upper bound

$$\max_{x \in [\varepsilon_0, \varepsilon_1]} \frac{\partial}{\partial x} h_{a,b}(x) \le O \left( \varepsilon \frac{m}{n} \log^2 n \right),$$

when $\frac{m}{n} - \sqrt{m/n}\log n \le a, b \le \frac{m}{n} + \sqrt{m/n}\log n$. Therefore, by the mean value theorem, we have that

$$|h_{a,b}(\varepsilon_1) - h_{a,b}(\varepsilon_0)| \le O\left(\varepsilon\delta\frac{m}{n}\log^2 n\right). \tag{23}$$

Following a similar argument, one can also derive the upper bound

$$|h_{a,b}(-\varepsilon_1) - h_{a,b}(-\varepsilon_0)| \le O\left(\varepsilon\delta\frac{m}{n}\log^2 n\right). \tag{24}$$

Combining Equations (18) to (20), (23) and (24) then gives that

$$|f_{a,b}(\varepsilon_1) - f_{a,b}(\varepsilon_0)| \le O\left(\varepsilon\delta\frac{m}{n}\log^2 n\right). \tag{25}$$

Recall that in the proof of Lemma B.1, we have that

$$\sum_{a,b\in\mathbb{N}} \frac{(\Pr(M_1 = a, M_2 = b \mid X = 0) - \Pr(M_1 = a, M_2 = b \mid X = 1))^2}{\Pr(M_1 = a, M_2 = b)}$$

$$= O(1) \sum_{a,b\in\mathbb{N}} \frac{1}{a!b!}\exp\left(-\frac{2m}{n}\right)\left(\frac{m}{n}\right)^{a+b}\frac{(f_{a,b}(\varepsilon_1) - f_{a,b}(\varepsilon_0))^2}{f_{a,b}(\varepsilon_1) + f_{a,b}(\varepsilon_0)}. \tag{26}$$

Hence, we proceed to bound from below $f_{a,b}(\varepsilon_0) + f_{a,b}(\varepsilon_1)$. Note that we have

$$f_{a,b}(\varepsilon_0) + f_{a,b}(\varepsilon_1) \ge (1 - x^2)^{\min(a,b)}(1 + x)^{|a-b|} \ge 1 - O(\min(a,b)x^2) \ge \Omega(1), \tag{27}$$

where the first inequality follows from a case analysis on whether $a > b$, in the second inequality we approximate $(1 - x^2)^{\min(a,b)}$ by $1 - O(\min(a,b)x^2)$ since $\min(a,b)x^2 = o(1)$ for $a, b$ in the assumed range and $x < \varepsilon$. Combining Equations (25) to (27) then gives that

$$\sum_{a,b\in\mathbb{N}} \frac{(\Pr(M_1 = a, M_2 = b \mid X = 0) - \Pr(M_1 = a, M_2 = b \mid X = 1))^2}{\Pr(M_1 = a, M_2 = b)}$$

$$\le O(1) \sum_{a,b\in\mathbb{N}} \mathbb{1}\left\{\frac{m}{n} - \sqrt{\frac{m}{n}}\log n \le a, b \le \frac{m}{n} + \sqrt{\frac{m}{n}}\log n\right\}\frac{1}{a!b!}\exp\left(-\frac{2m}{n}\right)\left(\frac{m}{n}\right)^{a+b}\varepsilon^2\delta^2\left(\frac{m}{n}\right)^2\log^4 n$$

$$+ O(1)\left(1 - \Pr_{a,b\sim\text{Poi}(m/n)}\left[m/n - \sqrt{m/n}\log n \le a, b \le m/n + \sqrt{m/n}\log n\right]\right)$$

$$\le \varepsilon^2\delta^2\left(\frac{m}{n}\right)^2\log^4 n + o(1/n).$$

This concludes the mutual information bound for the super-linear case.

$\square$

Lastly, we present the mutual information bound for the super-learning parameter regime, i.e., $m \ge \tilde{\Omega}\left(n/\varepsilon^2\right)$.

**Lemma B.4.** *Let $\delta = |\varepsilon_0 - \varepsilon_1|$ and $\varepsilon = \max(\varepsilon_0, \varepsilon_1)$. Let $m > \Omega\left(\frac{n}{\log^2 n\,\varepsilon^2}\right)$. Then it holds*

$$I(X : M_1, M_2) \le O\left(\frac{\varepsilon^2\delta^2 m^2\,\log^2 n}{n^2}\right).$$

We are going to use the following facts regarding KL-divergence and mutual information.

**Claim B.5** (Convexity of KL-divergence). *Let $w \in (0, 1)$, and $p_1, p_2, q_1, q_2$ be probability distributions over the same domain. Then it holds*

$$KL(wp_1 + (1 - w)p_2||wq_1 + (1 - w)q_2) \le wKL(p_1||q_1) + (1 - w)KL(p_2||q_2).$$

**Claim B.6** (Additivity of KL-divergence). *Let $w \in (0, 1)$, and $p_1, p_2, q_1, q_2$ be probability distributions such that $p_1, q_1$ share the same domain and $p_2, q_2$ share the same domain. Then it holds*

$$KL(p_1 \otimes p_2||q_1 \otimes q_2) = KL(p_1||q_1) + KL(p_2||q_2).$$

**Claim B.7** (Mutual Informtion Identity). *Let $X$ be an indicator variable, and $H_0, H_1$ be a pair of distributions. Let $H$ be the random variable that follows the distribution of $H_0$ if $X = 0$ and $X_1$ if $X = 1$. Then it holds*

$$I(X : H) = \frac{1}{2}KL\left(H_0 : \frac{1}{2}(H_1 + H_0)\right) + \frac{1}{2}KL\left(H_1 : \frac{1}{2}(H_1 + H_0)\right).$$

*Proof of Lemma B.4.* When $X = 0$, the pair $(M_1, M_2)$ is distributed as a even mixture of two distributions that are both product of Poisson distributions:

$$(M_1, M_2) \mid (X = 0) \sim H_0 := \frac{1}{2}\left(\text{Poi}\left(\frac{m}{n} + \varepsilon_0 \frac{m}{n}\right) \otimes \text{Poi}\left(\frac{m}{n} - \varepsilon_0 \frac{m}{n}\right) + \text{Poi}\left(\frac{m}{n} - \varepsilon_0 \frac{m}{n}\right) \otimes \text{Poi}\left(\frac{m}{n} + \varepsilon_0 \frac{m}{n}\right)\right).$$

Similarly, we have

$$(M_1, M_2) \mid (X = 1) \sim H_1 := \frac{1}{2}\left(\text{Poi}\left(\frac{m}{n} + \varepsilon_1 \frac{m}{n}\right) \otimes \text{Poi}\left(\frac{m}{n} - \varepsilon_1 \frac{m}{n}\right) + \text{Poi}\left(\frac{m}{n} - \varepsilon_1 \frac{m}{n}\right) \otimes \text{Poi}\left(\frac{m}{n} + \varepsilon_1 \frac{m}{n}\right)\right).$$

From Claim B.7 and Claim B.5, we know that it suffices to show that the KL-divergences between $H_0$ and $H_1$ are at most $\frac{\varepsilon^2 \delta^2 m^2 \log^2 n}{n^2}$. We focus on the argument for $\text{KL}(H_0 : H_1)$ as the one for $\text{KL}(H_1 : H_0)$ is symmetric.

By the definition of $H_0$ and $H_1$, we have that

$$\text{KL}(H_0 : H_1) \leq \frac{1}{2}\text{KL}\left(\text{Poi}\left(\frac{m}{n} + \varepsilon_0 \frac{m}{n}\right) \otimes \text{Poi}\left(\frac{m}{n} - \varepsilon_0 \frac{m}{n}\right) \middle\| \text{Poi}\left(\frac{m}{n} + \varepsilon_1 \frac{m}{n}\right) \otimes \text{Poi}\left(\frac{m}{n} - \varepsilon_1 \frac{m}{n}\right)\right)$$

$$+ \frac{1}{2}\text{KL}\left(\text{Poi}\left(\frac{m}{n} - \varepsilon_0 \frac{m}{n}\right) \otimes \text{Poi}\left(\frac{m}{n} + \varepsilon_0 \frac{m}{n}\right) \middle\| \text{Poi}\left(\frac{m}{n} - \varepsilon_1 \frac{m}{n}\right) \otimes \text{Poi}\left(\frac{m}{n} + \varepsilon_1 \frac{m}{n}\right).\right)$$

$$= \text{KL}\left(\text{Poi}\left(\frac{m}{n} + \varepsilon_0 \frac{m}{n}\right) \middle\| \text{Poi}\left(\frac{m}{n} + \varepsilon_1 \frac{m}{n}\right)\right) + \text{KL}\left(\text{Poi}\left(\frac{m}{n} - \varepsilon_0 \frac{m}{n}\right) \middle\| \text{Poi}\left(\frac{m}{n} - \varepsilon_1 \frac{m}{n}\right)\right),$$

where in the first line we use Claim B.5, and in the last line we use Claim B.6.

Let $\lambda_1 > \lambda_2 > 0$. Note that we have the following closed form for KL divergence between a pair of Poisson distributions:

$$\text{KL}(\text{Poi}(\lambda_1) \| \text{Poi}(\lambda_2)) = \lambda_1 \log(\lambda_1/\lambda_2) + \lambda_2 - \lambda_1 \leq \lambda_1 \frac{\lambda_1 - \lambda_2}{\lambda_2} + \lambda_2 - \lambda_1 = \frac{(\lambda_1 - \lambda_2)^2}{\lambda_2},$$

where in the first inequality we use the bound $\log(x) \leq x - 1$ for all $x > 0$. Hence, it follows that

$$\text{KL}(H_0 \| H_1) \leq \frac{\left(\frac{m}{n}(\varepsilon_1 - \varepsilon_0)\right)^2}{\frac{m}{n}(1 + \varepsilon_1)} \leq O(1)\frac{m}{n}\delta^2.$$

But notice that this is at most $O\left(\frac{\varepsilon^2 \delta^2 m^2 \log^2 n}{n^2}\right)$ as long as $m \geq \Omega\left(n/(2\log^2(n)\varepsilon^2)\right)$ which is our assumed parameter regime in this lemma. $\qquad\square$

By setting $\delta = |\varepsilon_1 - \varepsilon_0| = O(\varepsilon\rho)$, it follows from Lemmas B.1, B.3 and B.4 that $I(X : M_1, M_2) \leq O\left(\frac{\varepsilon^2 \delta^2 m^2 \log^4 n}{n^2}\right) + o(1/n)$. Since the pairs $M_i, M_{i+1}$ are conditional independent and have identical distributions given $X$, it follows that

$$I(X; M_1, \cdots, M_n) \leq \frac{n}{2} I(X; M_1, M_2) \leq O\left(\frac{\varepsilon^2 \delta^2 m^2 \log^4 n}{n}\right) + o(1).$$

This concludes the proof of Lemma 4.4.

## B.2  Proof of Lemma 4.3

*Proof.* Let $T$ be a set of samples. In case one, $T$ is made up of $m$ samples from a random distribution from $\mathcal{M}_0$. In case two, $T$ is made up of $m$ samples from a random distribution from $\mathcal{M}_1$. Assume that there exists a tester $\mathcal{A}$ that can distinguish between the two cases given $S$ with probability at least 0.9. We will show that this implies that $m = \tilde{\Omega}\left(\sqrt{n}\varepsilon^{-2}\rho^{-1}\right)$. Recall that Lemma 4.4 has the following setup. Let $X$ be an unbiased binary random variable. If $X = 0$, $S$ will be $\text{Poi}(2m)$

samples from a random distribution from $\mathcal{M}_0$. If $X = 1$, $S$ will be $\mathrm{Poi}\,(2m)$ samples from a random distribution from $\mathcal{M}_1$. Then we can use $\mathcal{A}$ to predict $X$ given $S$ in the following way: if $S$ contains more than $m$ samples, we feed $\mathcal{A}$ the first $m$ samples and take its output; otherwise, we declare failure. By the concentration property of Poisson distributions, it holds that $S$ contains more than $m$ samples with high constant probability. Hence, the above routine will be able to correctly predict $X$ with probability at least $0.6$. This implies that the mutual information between $X$ and $S$ is at least $\Omega(1)$. However, Lemma 4.4 says that the mutual information is at most

$$O\left(\varepsilon^4 \rho^2 \frac{m^2}{n}\,\log^4(n)\right) + o(1).$$

This therefore shows that

$$\varepsilon^4 \rho^2 \frac{m^2}{n}\,\log^4(n) = \Omega(1)$$

which further implies that

$$m = \Omega\left(\sqrt{n}\log^{-2}(n)\varepsilon^{-2}\rho^{-1}\right).$$

This concludes the proof of Lemma 4.3. $\qquad\square$

### B.3 Proof of Proposition 4.1

*Proof.* Let $m = \tilde{o}\left(\sqrt{n}\varepsilon^{-2}\rho^{-1}\right)$. For the sake of contradiction, assume that

$$|\mathrm{Acc}_m(\varepsilon_0) - \mathrm{Acc}_m(\varepsilon_1)| > 0.1$$

for some deterministic symmetric tester $\mathcal{A}(\,;r)$. Let $\mathcal{M}_0, \mathcal{M}_1$ be the local swap family of $\mathbf{p}(\varepsilon_0)$ and $\mathbf{p}(\varepsilon_1)$ respectively. We show that this could be used to distinguish between a random distribution from $\mathcal{M}_0$ and a random distribution from $\mathcal{M}_1$. This would then contradict Lemma 4.3. In particular, since the output of $\mathcal{A}(\,;r)$ is invariant up to domain relabeling, the acceptance probability of $\mathcal{A}(\,;r)$ when it takes samples from a random distribution from $\mathcal{M}_0$ must be exactly equal to $\mathrm{Acc}_m(\varepsilon_0)$ (and similarly for $\mathcal{M}_1$). This immediately implies a tester that takes $m$ samples and could distinguish between a random distribution from $\mathcal{M}_0$ and a random distribution from $\mathcal{M}_1$ with probability at least $1/2 + c$ for some small constant $c$. This further implies that if we takes $100mc^{-1}$ many samples, we can boost the success probability to $0.99$. Yet, since $100mc^{-1} = \tilde{o}\left(\sqrt{n}\varepsilon^{-2}\rho^{-1}\right)$, this clearly contradicts Lemma 4.3, and hence concludes the proof of Proposition 4.1. $\qquad\square$

### B.4 Proof of Theorem 1.6

*Proof.* The lower bound $\tilde{\Omega}(\varepsilon^{-2}\rho^{-2})$ follows from the fact that testing whether an unknown coin has bias $1/2$ or $1/2 + \varepsilon$ replicably requires $\tilde{\Omega}\left(\varepsilon^{-2}\rho^{-2}\right)$ many samples as shown in [30]. In the rest of the proof, we therefore focus on establishing the lower bound $\tilde{\Omega}(\sqrt{n}\varepsilon^{-2}\rho^{-1})$.

Let $\mathbf{p}(\xi)$ be the distribution instance defined as in Equation (4). Let $\mathcal{A}$ be a $\rho$-replicable uniformity tester that takes $m$ samples. We denote by $r$ the random string representing the internal randomness of $\mathcal{A}$.

Following the framework of [30], we will fix some random string $r$ such that (i) $\mathcal{A}(\,;r)$ accepts the uniform distribution with probability at least $1 - O(\rho)$ (ii) $\mathcal{A}(\,;r)$ rejects the distribution $\mathbf{p}(\varepsilon)$ with probability at least $1 - O(\rho)$ (iii) $\mathcal{A}(\,;r)$ is replicable with probability at least $1 - O(\rho)$ against $\mathbf{p}(\xi)$ for $\xi$ sampled uniformly from $[0, \varepsilon]$. By the correctness guarantees of $\mathcal{A}$, a large constant fraction of random strings $r$ must satisfy (i) and (ii). By the replicability requirement of $\mathcal{A}$, a large constant fraction of random strings $r$ must satisfy (iii). By the union bound, there must exist some random string $r$ that satisfies (i), (ii) and (iii) at the same time. Define the acceptance probability function

$$\mathrm{Acc}_m(\xi) = \Pr_{S \sim \mathbf{p}(\xi)^{\otimes m}}[\mathcal{A}(S;r) = 1].$$

Note that $\mathrm{Acc}_m(\xi)$ is a continuous function in $\xi$ since the acceptance probability can be expressed as a polynomial in $\xi$. Moreover, it holds that $\mathrm{Acc}_m(0) \geq 1 - O(\rho)$ and $\mathrm{Acc}_m(\varepsilon) \leq O(\rho)$. Therefore, there must exist some $\xi^*$ such that $\mathrm{Acc}_m(\xi^*) = 1/2$. Assume for the sake of contradiction that $m = \tilde{o}(\sqrt{n}\varepsilon^{-2}\rho^{-1})$. By Proposition 4.1, it follows that

$$\mathrm{Acc}_m(\xi) \in [\mathrm{Acc}_m(\xi^*) - 0.1, \mathrm{Acc}_m(\xi^*) + 0.1] = [0.4, 0.6]$$

for any $\xi$ such that $|\xi - \xi^*| \leq \rho\varepsilon$. In other words, if we sample $\xi$ randomly from $[0, \varepsilon]$, whenever $\xi$ falls into the interval $[\xi^* - \rho\varepsilon, \xi^* + \rho\varepsilon]$, the algorithm will fail to be replicable with constant probability. It is not hard to see that the interval has mass $\Omega(\rho)$ under the uniform distribution over $[0, \varepsilon]$. This would then imply that the tester would not be $O(\rho)$-replicable, contradicting property (iii). This shows that $m = \tilde{\Omega}(\sqrt{n}\varepsilon^{-2}\rho^{-1})$, and hence concludes the proof of Theorem 1.6. $\qquad\square$

## C    Further Discussion and Omitted Proofs

In this section we give additional discussions and omitted proofs.

### C.1    Barriers for $\chi^2$-Statistics

We show a similar barrier for using the $\chi^2$-test statistics. These statistics are used in the several uniformity testing algorithms, including [14, 1, 47, 20]. The $\chi^2$ test statistic (of [1] for example) computes

$$\sum_{i \in [n]} \frac{(X_i - m/n)^2 - X_i}{m/n}$$

where the algorithm takes $\mathrm{Poi}\,(m)$ samples and computes $X_i$ to be the frequency of the $i$-th element among the samples. For a uniform distribution, the test statistic is expected to be at most $m\varepsilon^2/500$, while if the test statistic is far from uniform the test statistic is at least $m\varepsilon^2/5$, leading to an expectation gap with around $m\varepsilon^2$ (Lemma 2 of [1]). Suppose a tester compares the test statistic with a random threshold sampled from the interval $[m\varepsilon^2/500, m\varepsilon^2/5]$. For the tester to be $O(\rho)$-replicable, we require that the test statistics deviate by no more than $O(\rho m\varepsilon^2)$ with high constant probability, in other words the variance must be at most $O(\rho^2 m^2 \varepsilon^4)$.

As before, we fix a constant $\varepsilon > 0$ so that we can ignore the dependency with sample complexity, and consider a hard instance of a distribution with a single heavy element with probability $p_i = n^{-1/2}$. In particular, we will have $X_i \sim \mathrm{Poi}\,(mp_i) = \mathrm{Poi}\,(m/\sqrt{n})$. Since $m/\sqrt{n} \gg 1$, there exists some constant small constant $c$ such that

$$\Pr[X_i > m/\sqrt{n} + c\sqrt{m}/n^{1/4}] > c\,,$$
$$\Pr[X_i < m/\sqrt{n} - c\sqrt{m}/n^{1/4}] > c.$$

However, in the two cases above, the contributions to the $\chi^2$-test statistic from $X_i$ differ by

$$\frac{\left(\frac{m}{\sqrt{n}} + c\frac{\sqrt{m}}{n^{1/4}} - \frac{m}{n}\right)^2 - \left(\frac{m}{\sqrt{n}} - c\frac{\sqrt{m}}{n^{1/4}} - \frac{m}{n}\right)^2 + 2c\frac{\sqrt{m}}{n^{1/4}}}{m/n} = \Omega\left(\frac{\frac{m^{3/2}}{n^{3/4}} + \frac{\sqrt{m}}{n^{1/4}}}{m/n}\right)$$
$$= \Omega\left(\sqrt{m}n^{1/4}\right).$$

Following our previous discussion, for the tester to be $O(\rho)$-replicable, we therefore require that $m^{1/2}n^{1/4} \ll \rho m$ or $m \geq n^{1/2}\rho^{-2}$.

### C.2    Discussion of Lower Bounds against Asymmetric Algorithms

For non-replicable uniformity testing algorithms, we can typically assume that symmetry is without loss of generality when analyzing the sample complexity. A common reduction that turns an asymmetric algorithm into a symmetric algorithm is the following. The algorithm could apply all permutations to the samples observed, and output the majority answer. However, it is not clear that such a transformation preserves replicability.

Below we discuss in more detail the barrier hit by our lower bound argument for asymmetric testers. Suppose we want to prove a lower bound against general algorithms. The known techniques for lower bounds against replicable algorithms first fix a random seed $r$ such that $\mathcal{A}(;r)$ is both correct and replicable with high probability given a distribution drawn from the adversarial instance (in our lower bound the instance draws distributions $\mathbf{p}(\xi)$ uniformly from $\xi \in [0, \varepsilon]$).[8] Given a fixed random seed,

---

[8]In our lower bound, we use the local swap family to restrict the adversary to a subset of permutations. In this discussion, we allow a more generic adversary that could permute the domain arbitrarily.

we now have a deterministic algorithm $\mathcal{A}(;r)$ that accepts $\mathbf{p}(0)$ and rejects any permutation of $\mathbf{p}(\varepsilon)$. By continuity, for any fixed permutation $\pi$, there is some $\xi_\pi$ such that $\mathcal{A}(;r)$ accepts with probability $1/2$ given samples drawn from $\mathbf{p}(\xi_\pi)$ permuted according to $\pi$. It remains to argue that algorithms with low sample complexity $m \ll \sqrt{n}\rho^{-1}\varepsilon^{-2}$ cannot successfully distinguish nearby distributions $\mathbf{p}(\xi_\pi), \mathbf{p}(\xi)$ (both permuted according to $\pi$) for any $\xi \in (\xi_\pi \pm \rho\varepsilon)$ so that the acceptance probability of $\mathcal{A}(;r)$ must be close to $1/2$ in this region and therefore $\mathcal{A}(;r)$ is not replicable with probability $\Omega(\rho)$. However, our arguments in Appendix B only show that the families $\mathbf{p}(\xi_\pi), \mathbf{p}(\xi)$ are indistinguishable when permuted by a random permutation $\pi'$, not necessarily when permuted by the permutation $\pi$. In fact our proof only allows us to argue that for any $\xi' \in [0, \varepsilon]$, the families are indistinguishable when permuted by a constant fraction of permutations $\pi'$. However, the permutations for which the families $\mathbf{p}(\xi'), \mathbf{p}(\xi' + \rho\varepsilon)$ are indistinguishable may not be the permutations for which $\xi_\pi, \xi'$ are close.

## C.3 Replicable Identity Testing

The formal definition of replicable identity testing is given below.

**Definition C.1** (Replicable Identity Testing). *Let $n \in \mathbb{Z}_+$, and $\varepsilon, \rho \in (0, 1/2)$. A randomized algorithm $\mathcal{A}$, given sample access to some distribution $\mathbf{p}$ on $[n]$, is said to solve $(n, \varepsilon, \rho)$- replicable identity testing if $\mathcal{A}$ is $\rho$-replicable and it satisfies the following for every fixed distribution $X$:*

*1. If $\mathbf{p}$ is $X$, $\mathcal{A}$ should accept with probability at least $1 - \rho$.* [9]

*2. If $\mathbf{p}$ is $\varepsilon$-far from $X$ in TV distance, $\mathcal{A}$ should reject with probability at least $1 - \rho$.*

[27] gives a reduction following reduction between uniformity and identity testing.

**Theorem C.2** ([27], restated). *Let $\mathbf{q}$ be a distribution over $[n]$ with known explicit description. There is a (randomized) algorithm $\mathcal{T}$ such that given $m$ independent samples to a distribution $\mathbf{p}$ on $[n]$ and $\varepsilon > 0$, generates $m$ independent samples from a distribution $\mathbf{p}'$ on $[6n]$ such that:*

*1. If $\mathbf{p} = \mathbf{q}$, $\mathbf{p}'$ is uniform on $[6n]$.*

*2. If $\mathbf{p}$ is $\varepsilon$-far from $\mathbf{q}$ in total variation distance, then $\mathbf{p}'$ is $\varepsilon/3$-far from uniform in total variation distance.*

*Furthermore, this algorithm runs in $O(m)$ time.*

If we want to design a replicable identity tester, we can simply transform the samples using $\mathcal{T}$ from Theorem C.2 (note that we don't even need to share the randomness of $\mathcal{T}$ across different runs) and then feed the transformed dataset to the replicable uniformity tester from Theorem 1.3. The correctness guarantees follow immediately. Let $\mathbf{p}'$ be the output distribution of $\mathcal{T}$. Since in both runs the replicable uniformity tester takes samples from $\mathbf{p}'$, replicability of the process follows from Theorem 1.3. Lastly, it is not hard to see that the number of samples consumed is asymptotically the same as the uniformity tester.

---

[9] As in the case of uniformity testing, we do not focus on the dependence of the sample complexity with the error parameter.

