# OpenReview forum: "Replicable Uniformity Testing"
_NeurIPS.cc/2024/Conference — NeurIPS 2024 poster_

### Official Review · Reviewer_g6cx · 2024-06-22

**Soundness:** 4
**Presentation:** 4
**Contribution:** 3
**Rating:** 7
**Confidence:** 4

**Summary:**

In this paper the authors study the problem of replicable uniformity testing. The non-replicable version of the problem can be stated as follows: given some $\varepsilon > 0$ and sample access to some distribution $p$ on $[n]$ what is the minimum number of samples $m$ to distinguish whether $p$ is the uniform distribution or $\varepsilon$ far from it in TV distance? A long line of work has shown that the sample complexity for this task is $\Theta(\sqrt{n}/\varepsilon^2)$. The authors ask for algorithms that have the additional replicability requirement based on the definition of Impagliazzo et al. (2022), meaning that when the algorithm is executed twice on two i.i.d. sets of samples from the same distribution it will make the same decision with probability at least $1-\rho$. The authors design an algorithm that requires $\tilde{\Theta}(\sqrt{n}/(\varepsilon^2\rho) + 1/(\varepsilon^2\rho^2))$ many samples from the distribution $p$. This bound is achieved by considering a (non-replicable) uniformity tester that is based on some $\ell_1$ statistic, since testers based on $\ell_2$ statistics would incur a $tilde{\Theta}(\sqrt{n}/(\varepsilon^2\rho^2))$ sample complexity. Moreover, they provide an (almost) matching lower bound for a natural class of "symmetric" algorithms.

**Strengths:**

I find this work pretty interesting. I think uniformity testing is a natural application domain for the definition of replicability, since the decisions of the algorithm are binary so the discrete metric on the output space (which is what the definition is asking for) is a natural one for this problem.

Moreover, the $\sqrt{n}/(\varepsilon^2 \rho)$ sample complexity is interesting and has not appeared in the replicability line of work before.

From a technical point of view, the upper bound is achieved by a standard modification of some non-replicable algorithm using a random thresholding trick. However the correctness is based on a concentration argument that requires subtle technical work, dependent on different regimes of the parameters that come into play.

Similarly, the lower bound follows the high-level template that has been introduced by prior work. Nevertheless, the technical details are not straightforward.

I think another aspect of the results that the authors could consider highlighting is that, in this setting, replicability does not require any blow-up in the sample complexity with respect to the ambient dimension. This is in contrast with other tasks such as replicable mean estimation. The straightforward generalization of the approach of [Impagliazzo, Lei, Pitassi, Sorrell '22] to the $\ell_\infty$ estimation of the means of $n$ coins requires a blow-up of $n^2/\rho^2$ samples. There is a (computationally inefficient) approach by [Karbasi, Velegkas, Yang, Zhou '23] that shaves off  a factor of $n$, and was recently shown to be optimal [Hopkins, Impagliazzo, Kane, Liu, Ye '24]. Similar results hold for the $\ell_2$ estimation.

Overall, I believe that the paper studies a useful problem, the results are interesting, and the technical contribution is above the bar for NeurIPS.

**Weaknesses:**

I think that the discussion prior to the statement of the main result could be improved by mentioning that there is a $1/(\varepsilon^2\rho^2)$ additive term in the sample complexity (maybe even in the discussion in the abstract).

I am bit confused about how the $\tilde{O}(1/(\varepsilon^2\rho^2))$ shows up in the upper bound. The proof sketch that is presented in the main body only considers the $\tilde{O}(\sqrt{n}/(varepsilon^2\rho))$ term. I understand that this is required even for $n=2$, I just don't see how it is used in the general case. Could you please elaborate on it?

**Questions:**

Please see the question in the weaknesses section.

Could you also please elaborate on the dependence of the sample complexity with respect to some error parameter? I'm referring to your discussion in footnote 1.

**Limitations:**

Addressed.

---

> ### Author Rebuttal · Authors · 2024-08-02
>
> We thank the reviewer for their thoughtful comments! In the proof sketch we have focused on the sub-linear regime ($m \leq n$). The dependence $1/(\varepsilon^{2} \rho^{2})$ is incurred to handle the super-learning regime ($m \geq n/\varepsilon^2$). We redirect the reviewer to the proof of Lemma 3.2 (Appendix A.1) for the analysis of this case. We will also add this dependence to the discussion in the main body.
>
> Our approach incurs a $\log (1/\delta)$ overhead in the sample complexity. In particular, for arbitrary error parameter $\delta$, we may replace $m_0 \gets \log (1/\rho)$ with $m_0 \gets \log(1/\min(\rho, \delta)) = \log(1/\delta)$ (assuming $\delta \leq \rho$). This guarantees that in the completeness and soundness regimes, the test statistic is sufficiently concentrated such that the random threshold will lie on the correct side of the empirical test statistic. Using our approach, this $\log(1/\delta)$ approach seems necessary, however, we do not know if this is tight. We hope to investigate in future work whether this dependence is necessary in general, or if better sample complexity (say $\sqrt{\log(1/\delta)}$) is possible.

---

> > ### Comment · Reviewer_g6cx · 2024-08-09
> >
> > Thank you for your response, after reading the rest of the reviews and your rebuttal I remain positive about the paper.

---

### Official Review · Reviewer_sPzn · 2024-07-12

**Soundness:** 4
**Presentation:** 3
**Contribution:** 4
**Rating:** 8
**Confidence:** 4

**Summary:**

This work studies uniformity testing in the context of replicable algorithms. Knon uniformity testing algorithms are non-replicable in the following sense:  a) If the unknown distribution equals the uniform, then they output 1 whp, b) if they are epsilon-away from uniform, output 0 whp c) when the distance is between 0 and epsilon, then they output 1 with arbitrary (could be 1/2) probability. Thus when (c) happens, two different runs of the algorithm may give two different answers, thus are not replicable.  The goal of this work is to design algorithms so that even when (c) happens, the algorithms are  \rho-replicable (jn the sense of ILPS22).  The obvious \rho-replicable algorithms will have a blow-up 1/\rho^2 in the sample complexity (compared to non-replicable algorithms). The main contribution of this work is to show that this can be achieved via only a 1/\rho-blowup in sample complexity. This work also shows that 1/\rho-bloup is necessary if the algorithms have certain property (permutation-invariant).

**Strengths:**

The cost of replicability (in terms of sample complexity) is not well understood. ILPS shows that a blow-up of 1/\rho^2 is needed when the sample space size in 2. I belive this is the only known result.  It is natural to expect that a similar lowerbound holds for high-dimensional distributions. However, rather surprisingly, this work shows that this is not necessarily the case. Further elucidating the point hat the tradeoff between sample complexity and replicability needs much further investigation by the community. I really like the contribution of the work.

**Weaknesses:**

I am hoping to uncover a high-level explanation as  why the dependency on \rho is inverse-linear when $n$ is high.  The additive term has 1/\rho^2 term which is consistent with the lowerbound of ILPS.  To me, this suggests that there is an intuitive explanation of the sample complexity, however, I am not able to uncover such an explanation. The overview that the authors provide is more technical than intuitive.

**Questions:**

1. Related to weakness. Can you the high-level intuition?

---

> ### Author Rebuttal · Authors · 2024-08-02
>
> We thank the reviewer for the encouraging comments. Below we provide some more succinct explanation of this surprising linear dependency on $\rho$. First we note that this sample complexity is perhaps not so surprising if one focuses on designing testers for the specific hard instance we constructed in our lower bound section. In particular, for constant $\varepsilon$, the i-th element of the distribution has probability mass of the form $( \pm  \xi) / n $, where $\xi$ will be some randomly sampled value bounded from above by some constant $\varepsilon$.
>
> In that case, it amounts to estimating the squared  $\ell_2$ norm of the distribution up to accuracy $\rho / n$ (which is simply $\rho$ times the usual expectation gap for uniformity testing). For this particular instance, since there are no elements whose mass is heavier than $2 / n$, the variance of the usual collision statistics is at most of order $m^2 / n$. Solving $m^2 / n < \sqrt{\rho / n}$ then gives the sample complexity bound of $m = \Theta( \sqrt{n} / \rho )$. In short, at least in this case, the reason we have this surprising linear dependency on $\rho$ is conceptually similar to the birthday paradox phenomenon (where we expect an $n$ dependency for uniformity testing but $\sqrt{n}$ turns out to be sufficient).
>
> The main technical challenge is to analyze the case when there are heavy elements.
> While the presence of heavy elements makes collision-based test statistics sub-optimal (as discussed in the technical overview), it turns out that their presence quickly pushes the expected value of the TV test statistics to the soundness regime, leading to replicable rejection.

---

> > ### Comment · Reviewer_sPzn · 2024-08-09
> >
> > Thank you for the response.

---

### Official Review · Reviewer_MK1R · 2024-07-13

**Soundness:** 1
**Presentation:** 1
**Contribution:** 1
**Rating:** 4
**Confidence:** 5

**Summary:**

The concept of "reproducible" learning was introduced in STOC 22 paper and the concept is a very relevant to modern day research.
They also showed how algorithms based on statistical estimations can be easily converted to reproducible algorithm with a little overhead.
In this paper the authors have tried to produce reproducible testing algorithm for uniformity testing.  Uniformity is indeed a very important problem is a lot of applications.

**Strengths:**

NA

**Weaknesses:**

There are many algorithms for uniformity testing. Many of them are either already reproducible or since they are based on statistical estimations they can be easily converted into reproducible algorithms. Neither has the literature of the uniformity testing done thoroughly, nor has it been discussed why this algorithm is new. The same, or similar, algorithm has already been used in the literature.

**Questions:**

NA

---

> ### Author Rebuttal · Authors · 2024-08-02
>
> We thank the reviewer for their comments. While we have reviewed the extensive line of work on uniformity testing to the best of our abilities, we admit that our survey might not be complete. Thus, we would greatly appreciate any pointers to missing citations.
>
> Indeed, there is an extensive literature on uniformity testing, with a variety of algorithmic techniques. However, none of these algorithms explicitly guarantee replicability when the input distribution is neither uniform nor far from uniform. While there is a simple transformation to obtain replicable algorithms (by treating the outcome of a non-replicable algorithm as a coin flip), any such reduction introduces quadratic overhead in the replicability parameter. In particular, these algorithms require $\sqrt{n} \rho^{-2} \varepsilon^{-2}$ samples.
>
> Our key technical contribution is a replicable algorithm that achieves sample complexity $\sqrt{n} \rho^{-1} \varepsilon^{-2}$ in the worst case, and is thus the first algorithm (as far as we know for any statistical task, not just uniformity testing) that has linear overhead in the replicability parameter. While the algorithm is a simple modification of known algorithms, our primary technical contributions are new analyses in the concentration of the test statistic. Furthermore, as discussed in the technical overview, it is not immediately clear which of the many uniformity algorithms can be made replicable with linear overhead. We find one such test statistic that admits a replicable variant, and introduce new analysis to show that this is the case.

---

> > ### Comment · Reviewer_MK1R · 2024-08-13
> >
> > Thanks for the detailed response.

---

### Official Review · Reviewer_H8kM · 2024-07-15

**Soundness:** 3
**Presentation:** 3
**Contribution:** 2
**Rating:** 5
**Confidence:** 3

**Summary:**

This paper studies uniformity testing under the replicability constraint. Given samples from an unknown distribution P, we need to decide whether P is uniform or eps far from uniform with high probability. Additionally, the algorithm needs to report the same answer in two different random input samples with high probability.

The main contribution is an algorithm that takes 1/rho factor more samples than the non-replicable counterpart for which tight sample complexity is well-known. When the sample space size is 2, a 1/rho^2-factor blowup is known to be necessary from prior work. Thus, the new algorithm shaves off a rho factor. Matching sample complexity lower bound is shown only under symmetric assumptions where the algorithm behaves identically under a renaming of the sample space.

**Strengths:**

The main strength of the paper is the improved upper bound. At this point several different approaches for the non-replicable uniformity testing are well-known. The authors manage to show that the empirical TV approach in fact gives the best bounds under replicability constraints.

**Weaknesses:**

The main weakness I believe is the lack of an unconditional tight sample-complexity lower bound.

**Questions:**

None.

**Limitations:**

None.

---

> ### Author Rebuttal · Authors · 2024-08-02
>
> While our lower bound holds only against symmetric algorithms, we remark that all known uniformity testers in prior works are indeed symmetric. Moreover, in our opinion, symmetric algorithms are natural for the problem of uniformity testing as the property itself is invariant under domain relabeling. It is unclear whether there is any natural/intuitive way to exploit asymmetry in replicable algorithm design. We believe the fact that our lower bound holds only against symmetric algorithms is more of a technical issue than a conceptual gap, and we leave it as an important future direction to develop a fully unconditional lower bound.

---

> > ### Comment · Reviewer_H8kM · 2024-08-14
> > **Read the rebuttal**
> >
> > Thanks for the reply from the authors. I have not found enough evidence to change my assessment of the paper.

---

### Decision · Program_Chairs · 2024-09-25

**Decision:**

Accept (poster)

**Comment:**

The paper gives a new analysis of a uniformity test under replicability constraints, showing smaller blow up over the non-replicable setting than prior work, and a lower bound against symmetric testers that nearly matches the upper bound. The result is interesting and novel, and addresses a basic problem in a model that is new but has received a lot of attention recently.